



# Mixing at the extratropical tropopause as characterized by collocated airborne $H_2O$ and $O_3$ lidar observations

Andreas Schäfler[1], Andreas Fix[1], Martin Wirth[1]

[1]Deutsches Zentrum für Luft- und Raumfahrt, Institut für Physik der Atmosphäre, Oberpfaffenhofen, Germany

*Correspondence to*: Andreas Schäfler (Andreas.Schaefler@dlr.de)

**Abstract**

The composition of the extratropical transition layer (ExTL), which is the transition zone between the stratosphere and the troposphere in the mid-latitudes, largely depends on dynamical processes fostering the exchange of air masses. Here we follow the need to better characterize the ExTL in relation to the dynamic situation using the first-ever collocated airborne

lidar observations of ozone ($O_3$) and water vapour ($H_2O$) across the tropopause. The potential of such lidar profile data, required for a novel, two-dimensional depiction of the complex trace gas distributions and mixing along cross-sections, is illustrated for a perpendicular jet stream crossing during a research flight over the North Atlantic conducted on 1 October 2017 in the framework of the Wave-driven Isentropic Exchange (WISE) field campaign.

The analysis of the ExTL shape and composition uses a combined view of the lidar data in geometrical and tracer-tracer (T-

T) space, which was so far not possible from existing observations. For this particular case study, which is considered to be representative for the climatological distribution, the T-T depiction allows to identify distinct mixing regimes that suggest mixing of air masses with differing origin: we find clearly separated mixing of stratospheric air with moist extratropical air as well as with dry tropical air in the surrounding of the extratropical jet stream. This separation is indicative for differing transport pathways in the troposphere which need to be further elaborated using Lagrangian diagnostics. The $O_3$ and $H_2O$

distributions confirm strongest mixing above and below the maximum jet stream winds, while it is suppressed in-between. The interrelation of chemical and dynamical discontinuities is investigated and strongest isentropic trace gas gradients are found to be better correlated with maximum isentropic PV gradients than with classical dynamical tropopause definitions. Although the methods neither allow conclusions on the individual mixing process nor on the location and time of the event, the consideration of data subsets allows discussing the formation and interpretation of isentropic and vertical mixing lines in

T-T space and to develop hypotheses on mixing at different time-scales. The presented two-dimensional lidar data is considered to be of relevance for the investigation of further synoptic situations leading to mixing across the tropopause and for future validation of chemistry and numerical weather prediction models.



# 1 Introduction

The extratropical transition layer (ExTL) is a subregion of the extratropical upper troposphere and lower stratosphere
(ExUTLS) which is relevant both for climate (Riese et al. 2012) and weather (Gray et al., 2014). Radiatively active trace
gases, like ozone ($O_3$) and water vapour ($H_2O$) provide significant vertical gradients across the ExTL, and the tropopause
therein, that impact the earth's radiation budget. The transition from the troposphere to the stratosphere can be abrupt or
more uniform in cases where the ExTL is strongly impacted by two-way stratosphere-troposphere exchange (STE)
processes. Depending on their life time, observed trace gases reveal a footprint of mixing processes in their Lagrangian
history, typically as an intermediate chemical characteristic with both tropospheric and stratospheric influence, highlighting
irreversible and bidirectional transport between the spheres (Gettelman et al. 2011). These mixing processes strongly depend
on the dynamical situation. For a better understanding of the role of multi-scale dynamical processes on the composition of
the ExTL in the mid-latitudes the Wave-driven Isentropic Exchange (WISE) field campaign (Kunkel et al., 2019) was
conducted over the North Atlantic Ocean in autumn 2017. The HALO (High Altitude LOng range) research aircraft
performed in situ and remote sensing measurements of various trace gases in the ExTL from turbulence to synoptic scale in a
variety of synoptic situations.

Mixing in the extratropics is often related to upper-level frontal zone–jet stream systems (Keyser and Shapiro, 1986; Lang
and Martin, 2012) that are characterized by isentropic surfaces that cross the sloped tropopause (Holton et al., 1995; Stohl et
al., 2003). The highly variable mid-latitude flow is largely affected by baroclinic cyclones that develop from disturbances in
the jet stream and cause a strong distortion of the tropopause through redistribution of tropospheric and stratospheric air
masses. Intrusions of stratospheric air into the troposphere are connected to jet streams and cyclones and represent areas of
irreversible mixing of tropospheric and stratospheric air due to filamentation (Danielsen, 1968; Danielsen et al., 1987) and
roll-up up of intrusions (Appenzeller et al., 1996). Strong wind shear above and below the jet stream maximum results in
clear air turbulence fostering the exchange between stratosphere and troposphere (Shapiro, 1976; Shapiro, 1980). Recently,
Spreitzer et al. (2019) have shown the importance of turbulence in upper-level frontal zone–jet stream systems and
tropopause folds. Beside turbulence, a variety of other non-conservative diabatic processes occur near jet streams and
cyclones that foster cross-isentropic mixing, e.g., cloud diabatic processes in convective or large-scale clouds (Gray et al.,
2003; Wernli and Bourqui, 2000) or radiative processes related to vertical $H_2O$ gradients or clouds (Zierl and Wirth, 1997).
Additionally, thunderstorms were shown to impact the ExTL composition (e.g., Huntrieser et al., 2016; Pan et al., 2014a)
often being triggered by large-scale weather systems. The spatiotemporal diversity of the flow and the complex life of
cyclones results in a large variety of mixing and exchange processes that were found from case studies and climatologies



(Sprenger et al., 2003; Škerlak et al., 2014; Reutter et al., 2015; Boothe et al., 2017) and explains the complexity that ExTL observations have shown in terms of their chemical characteristics.

Mixed air masses can be identified by relationships between long-lived chemical trace gases (Hintsa et al., 1998, Fischer et al., 2000; Hoor et al., 2002; Zahn and Brenninkmeijer, 2003; Pan et al., 2004). This correlation of tropospheric and stratospheric tracers with opposing behaviour (tracer-tracer or T-T correlation, explained in more detail in Sect. 2.2), e.g., of $O_3$ and $H_2O$, was used to separate mixed air masses of intermediate chemical characteristics from the undisturbed background to explore the average composition and extent of the ExTL (see summary in Gettelman et al., 2011). Many of these climatological studies made use of data from multiple research flights, multiple campaigns or used satellite data. In situ observations of chemical species on board commercial aircraft (e.g., Brenninkmeijer et al., 2007) are restricted to the flight routes and the altitude range of the aircraft and provide only a limited number of vertical profiles during start and landing (Zahn et al., 2014). The use of satellite observations guarantees a high temporal resolution and global coverage, however, is limited in vertical resolution (about 1–3 km in the UTLS) and rather high measurement uncertainty in the tropopause region (Hegglin et al., 2008; Hegglin et al., 2009). Aircraft in situ data obtained during research campaigns are highly accurate and temporally resolved, however, with limited spatial and temporal coverage.

Several case studies, typically using repeated in situ flight legs at different altitudes to provide a certain altitude resolution showed a strong influence of the synoptic situation on the interplay of dynamics and chemistry (e.g., Pan et al., 2007; Vogel et al., 2011; Konopka and Pan, 2012). Pan et al. (2007) contrast two different dynamical conditions, a strong jet stream with a complex tropopause fold structure and a flat tropopause situation, and found a correlation between the sharpness of the chemical and thermal transitions with minimal mixing in the flat tropopause situation. Mixed air masses dominated on the cyclonic jet stream side in an area where the dynamical and thermal tropopause were separated. Konopka and Pan (2012) used in situ observations in combination with a trajectory model to demonstrate that large parts of the ExTL near a jet stream are formed on time scales of a few days, especially in the lower part of the jet stream.

For a more detailed characterization of the influence of the dynamics in individual synoptic situations on the distribution of trace gases in the ExTL, an observation capability for the instantaneous two-dimensional distribution of relevant trace gas along cross-sections through individual weather systems is required. Two-dimensional profiles from active and passive remote sensing instruments on board research aircraft can fill this observational gap between airborne in situ and satellite measurements. Passive airborne limb sounders allowed for retrieving vertical profiles of a multitude of trace gas species near the subtropical jet stream (Ungermann et al., 2013). Limb sounder instruments provide a good along track (~3 km) and vertical resolution (200 to 300 m depending on the observed altitude) to resolve tropopause-based gradients. However, the





low line-of-sight resolution requires homogeneity in viewing direction as gradients can cause artefacts in the trace gas profiles. Woiwode et al. (2019) investigated the applicability of the linear limb-imaging GLORIA (Gimballed Limb Observer for Radiance Imaging of the Atmosphere) to observe the fine structure of a tropopause fold. Active remote sensing with airborne Differential Absorption Lidar (DIAL) offers both, a high horizontal and vertical resolution directly beneath the

90 aircraft. Early pioneering studies demonstrated the significance of range-resolved profiles of $O_3$ (Browell et al., 1987) and of $H_2O$ (Ehret et al., 1999) to characterize mesoscale tropopause folds. The benefit of using simultaneous lidar measurements of $H_2O$ and $O_3$ was emphasized by Kooi et al. (2008) showing observations in the tropical troposphere. However, their used DIAL is not capable to accurately measure the low $H_2O$ mixing ratios occurring in the stratosphere (Browell et al., 1998). During WISE, the first-ever collocated DIAL nadir profile observations of $H_2O$ and $O_3$ (Fix et al., 2019) were made in the

95 ExTL across the tropopause, which offers instantaneous information on the structure of the ExTL in dependence of the underlying dynamical situation.

In this paper we describe the DIAL $O_3$ and $H_2O$ lidar observations (Sect. 2.1) and their combined application to established T-T diagnostics (Sect. 2.2). This provides a novel way to gain information about the ExTL mixing state along a cross-section using a back projection of information from T-T space to geometrical space, which was so far not possible from existing

100 observational capabilities. The aim is to test whether the curtain-like observations of $O_3$ and $H_2O$ across the tropopause are sufficient to characterize the complex chemical structure in the ExTL for a text-book situation with a transect of a zonal jet stream observed over the North Atlantic Ocean during WISE on 1 October 2017 (Sect. 3.1). For the first time, distinct mixing regimes for a range of isentropic levels across the tropopause allow a detailed depiction and description of the transition from stratosphere to troposphere (Sect. 3.2–3.4). Additionally, isentropic $O_3$ and $H_2O$ gradients are determined to

105 investigate the interrelation of chemical and dynamical discontinuities at the mid-latitude tropopause (Sect. 3.5). Section 4 gives a summary of the results. In a follow-up paper we aim to apply Lagrangian diagnostics to investigate the role of differing transport pathways and the timescales that were relevant for the complex distribution of $O_3$ and $H_2O$ in this case.

## 2 Data and methods

### 2.1 Lidar observations onboard HALO

110 During the WISE campaign, the German research aircraft HALO (Krautstrunk and Giez, 2012) was equipped with the Water Vapour Differential Absorption Lidar in Space (WALES), which was originally designed as a four-wavelength $H_2O$ DIAL operating at 935 nm (Wirth et al., 2009). In the past, WALES was characterized and applied in multiple campaigns focussing on various topics ranging from atmospheric dynamics (e.g. Schäfler et al., 2015; Schäfler et al., 2018), moisture transport



(e.g., Schäfler et al., 2010; Kiemle et al., 2011), cloud-microphysics (Urbanek et al., 2017) to UTLS investigations (Trickl et al., 2016). In 2012, the system was extended by an optional $O_3$-DIAL capability (Fix et al., 2019) to be able to measure collocated profiles of $O_3$ and $H_2O$. During the "Polar Stratosphere in a Changing Climate" (POLSTRACC; Oelhaf et al., 2019) campaign in 2016, this capability was used for the first time, however, in a zenith pointing mode for stratospheric observations.

The measurement principle of the DIAL is based on the differential absorption of laser pulses at two or more wavelengths. The spectrally close wavelengths are selected such that absorption and scattering properties on their way through the atmosphere only differ with respect to absorption by the trace gas of interest, i.e. $O_3$ and $H_2O$. Accordingly, the system creates two wavelengths for $H_2O$ DIAL in the absorption band at 935 nm and two wavelengths for $O_3$ DIAL at 305 and 315 nm. For both pairs of wavelengths, one wavelength provides a strong absorption depending on the trace gas concentration, while the other is absorbed only weakly resulting in a stronger backscatter signal. From the ratio of both signals in dependence of the time needed for passing the atmosphere and the knowledge about the exact absorption characteristics, a range-dependent determination of $O_3$ and $H_2O$ number densities in the illuminated volume becomes possible. To reduce statistical noise in the signals, these are temporally averaged over 24 s, which corresponds to a 5.6 km distance between neighbouring profiles. In this study, $O_3$ and $H_2O$ is determined every 15 m in the vertical although it has to be mentioned that the effective vertical resolution of the data is 500 m (full width at half maximum (FWHM) of the averaging kernel) and exactly the same for $O_3$ and $H_2O$. The observed number density from the DIAL is converted to volume mixing ratios (VMR) using profiles of temperature and pressure typically taken from numerical weather prediction models (see Sect. 2.3). For a detailed characterization and validation of the instrument, the interested reader is referred to Wirth et al. (2009) and Fix et al. (2019). Note that throughout this study $H_2O$ VMR is given as ppm which is equivalent to $10^{-6}$ mol mol$^{-1}$ or μmol mol$^{-1}$ and $O_3$ VMR uses ppb which is equivalent to $10^{-9}$ mol mol$^{-1}$ or nmol mol$^{-1}$.

## 2.2 Tracer-tracer correlation

One of the key methods that is applied here is the presentation of the lidar data in T-T space, which is a well-established method to investigate the chemical transition in the ExTL (Hintsa et al., 1998, Fischer et al., 2000, Zahn and Brenninkmeijer, 2003). When the concentration of a trace gas with its main sources in the stratosphere is displayed in relation to the concentration of another trace gas with its main sources in the troposphere, in the idealized situation of no mixing, this T-T correlation method shows a L-shaped distribution with two characteristic branches of nearly linear relationships for the tropospheric and the stratospheric branch, respectively (e.g. Hoor et al., 2002; Pan et al., 2004). Such L-shaped distributions,





in case of $H_2O$ and $O_3$, typically occur in the tropics, where cross-tropopause mixing is weak and where slowly ascending tropospheric air masses are efficiently dehydrated at the cold tropical tropopause (Hegglin et al., 2009; Pan et al., 2014b; Pan et al., 2018). In the midlatitudes, where many of the above listed STE processes occur, observations show transition states

aligned along so-called "mixing lines" between the two branches, which represent a chemical signature from the stratosphere and the troposphere and connect both. The slope of the linear mixing lines critically depends on the concentration of the initial air masses in troposphere and stratosphere that are involved in the mixing process (Hoor et al., 2002). Photochemistry may lead to curved mixing lines (Hoor et al., 2002).

Several studies analysed the depth and composition of the ExTL displaying multi-flight in situ data sets in T-T space (see

Sect. 1) which showed compact regions of mixing lines in T-T space that allowed to delineate the mixing layer (e.g. Pan et al., 2007). The compactness of the mixing lines may be explained by the rather weak variability of the tropospheric and stratospheric trace gas concentrations that are connected by the mixing lines on time scales of individual research campaigns (Hegglin et al., 2009). This allowed a statistical investigation of the ExTL, although the individual flights may have covered various dynamical situations and air masses of different origin. Pan et al. (2004) found increased ExTL sharpness with

increasing latitude and the ExTL being centred on the thermal tropopause. Hoor et al. (2004) showed that the influence of STE decreases with horizontal and vertical distance from the tropopause and that the ExTL closely follows the tropopause. They also found a weak seasonality in STE with the ExTL reaching up to 30 K above the local tropopause in summer when isentropic PV gradients are weak and STE is intensified. The climatological analysis by Hegglin et al. (2009) shows an increase of ExTL depth above the thermal tropopause with latitude, increasing from 1–1.5 km in the subtropics to 3–4 km in

the northern hemispheric polar regions. Pan et al. (2007) found that the choice of the tropopause definition results in a different interpretation of the STE influence on the troposphere and stratosphere.

Hoor et al. (2004) found distinct mixing lines for particular flights as a result of individual synoptic situations. Pan et al. (2014b) mention that a combined analysis in T-T and geometric space with dynamical information can provide the information required to identify the location of the transition in relation to the dynamical structure associated with the air

mass boundary. This combined approach was used to locate mixed, stratospheric and tropospheric air masses along selected flight legs crossing the ExTL horizontally and vertically (Pan et al., 2007; Vogel et al., 2011; Konopka and Pan, 2012). However, a detailed attribution of mixing lines to locations in geometrical space and an isentropic investigation was so far limited by the reduced information content of staggered in situ legs.

A prerequisite for the T-T method is that the distributions are controlled by transport processes i.e. that the lifetime of the

used trace gases is longer than the timescale of transport and mixing at the tropopause, which is in the order of weeks.



Different trace gases were used; for example, carbon monoxide (CO) as tropospheric tracer in combination with $O_3$ as stratospheric tracer (e.g. Hoor et al., 2002). Pan et al. (2007) discuss the applicability of $O_3$–$H_2O$ correlations and note that $H_2O$ is a suitable tropospheric tracer despite the fact that it is not perfectly long-lived as phase changes may cause non-conservation of the gas phase $H_2O$ concentration. As discussed in Hegglin et al. (2009), the exponential decrease of $H_2O$

across the tropopause makes it a very useful source of information about transport into the stratosphere as even small amounts of $H_2O$ become visible as signature of increased $H_2O$. In the stratosphere, methane oxidation can produce $H_2O$ which is, however, rather small in the LS and therefore often neglected (Pan et al., 2014b). Due to the large dynamic range of $H_2O$ of four orders of magnitude from the troposphere to the stratosphere, T-T depictions use a logarithmic scaling for $H_2O$ to be able to distinguish the mixing lines (e.g., Hegglin et al., 2009; Tilmes et al., 2010), which are typically curved in

lin-log T-T diagrams, more easily.

Stratospheric and tropospheric background distributions are usually selected in T-T space by defining case-dependent threshold concentrations. One method uses thresholds e.g. $H_2O \leq 5$ ppm to select the stratospheric branch and $O_3 \leq 65$ ppb for the tropospheric branch in combinations with linear fits to the selected data in the two branches (Pan et al., 2004; Pan et al., 2007). Data points outside the $2\sigma$ level of both branches are considered to be mixed air masses. In a more simple

approach, Pan et al. (2014b) used probability density functions of the observations to separate undisturbed background from mixed air masses. The choice of the thresholds for the background distributions and the used combination of trace gases may impact the determination of the ExTL depth and also depend on the data set in terms of where and when during the year it was obtained (Hegglin et al., 2009; Tilmes et al., 2010). Woiwode et al. (2019) used two-dimensional passive remote sensing data with fixed thresholds to determine mixed air masses from T-T correlations of $H_2O$ and $O_3$, however, did not give further

detail about the composition of the ExTL. In this study the selection is done using a combined view of T-T and geometrical distributions which is only possible from the collocated lidar measurements to come as close as possible to a correct identification of mixed, stratospheric and tropospheric air masses.

### 2.3 Meteorological data

Unfortunately, no collocated profile observations of wind and temperature are available providing a similar resolution and

coverage as the DIAL data. In order to put the observational data in the context of the dynamical situation we use one-hourly meteorological reanalysis fields from the European Centre for Medium-Range Weather Forecasts (ECMWF) ERA-5 data set (Hersbach et al., 2020) retrieved on a 0.5/0.5 degree grid with 137 vertical levels. The reanalysis fields were interpolated bilinearly in space and linearly in time towards the observation location (Schäfler et al., 2010). Note that the vertical


separation of model levels in the tropopause region is about 300 m (e.g. Schäfler et al., 2020). The main parameters of interest are wind speed to identify the jet stream, potential vorticity for the location of the dynamical tropopause as well as pressure and potential temperature for the vertical context. Although analyses from numerical weather prediction models have significantly improved in the past, it is well-known that dynamic and thermodynamic quantities show uncertainties especially in regions of strong vertical gradients, i.e. the tropopause (e.g. Schäfler et al. 2020). However, it is deemed sufficient to provide the large-scale dynamical context that is relevant to interpret the observations. Additionally, error

sources resulting from temporal and spatial interpolation are neglected.

## 3 O$_3$ and H$_2$O observations on 1 October 2017

### 3.1 The synoptic setting

During the WISE campaign, a total of 17 flights were conducted over the eastern North Atlantic Ocean and western Europe with HALO out of Shannon, Ireland between 13 September and 21 October 2017. Collocated O$_3$ and H$_2$O DIAL

measurements were made in a number of meteorological situations including several crossings of extratropical jet streams and tropopause folds, multiple high-altitude observations of warm conveyor belts (WCB) outflows, i.e. strongly ascending tropospheric air streams reaching the tropopause (Browning et al., 1973; Wernli and Davies, 1997) and several crossings of filamentary structures in occluded frontal systems; i.e. typical phenomena related to breaking Rossby waves. A case study on in situ observations above a WCB outflow is discussed in Kunkel et al. (2019).

Here we focus on one particular case on 1 October 2017 that was characterized by a straight south-westerly jet stream over the North Atlantic Ocean (Fig. 1a) located between a large-scale longwave trough over the western Atlantic and a ridge extending over the North Sea into Southern Scandinavia (Fig. 1b). Two surface cyclones evolved in the upstream trough (L1 and L3 in Fig. 1) that feature typical cyclonic cloud patterns at 400 hPa, which are indicative for vertical transport of tropospheric air ahead of the trough. Further downstream, a surface cyclone was located between the UK and Iceland (L2)

and another surface low (L4) is visible over the central North Atlantic south of the strong jet stream. The 350 K potential vorticity (PV) distribution intersects with the jet stream that follows the maximum gradient in PV (Martius et al. 2010) and separates stratospheric air (> 2 Potential Vorticity Units (PVU = $10^{-6}$ K m$^2$ kg$^{-1}$ s$^{-1}$)) north of the wind speed maximum from tropospheric (< 2 PVU) to its south. In this region of strong horizontal and vertical velocity gradients and neighboring air masses of different origin, isentropic mixing was expected to have influenced the ExTL. The synoptic pattern was found to

be relatively persistent over the preceding hours. Stratospheric air was transported all around the subtropical anticyclone


keeping its high levels of PV in contrast to the tropospheric low-PV air that was advected northeastward on the leading edge of the upstream longwave trough filling the center of the anticyclone (Fig. 1a).

HALO performed a flight between 12:04 and 21:57 UTC that aimed at observing predicted strong tracer gradients and mixing across the jet stream using both, in situ and remote sensing instrumentation. Multiple, almost perpendicular crossings

of the jet stream at 13° W and 15° W were performed along a rectangular-shaped flight pattern at different altitudes for in situ (FL 280, FL 410, FL 430) and remote sensing measurements (FL 450). In this study, we concentrate on the last jet stream transect at 13° W from 18:40 to 20:00 UTC (green line in Fig. 1) only, providing maximum data coverage beneath the aircraft at FL 450. As this study focusses on processes in the jet stream, the northern- and the southernmost part of the zonal flight track were omitted. In those parts of the flight, either air mass transport in the occluded cyclone L2 or older

stratospheric air that was advected around the upper-level anticyclone (Fig. 1a) were encountered which are not relevant for this study. On the considered transect between 50.5° N and 60.5° N HALO crossed a zonally extended cloud band related to cyclone L4 (see Fig. 1b) which coincides with the jet axis on the northern side of the clouds (Fig. 2).

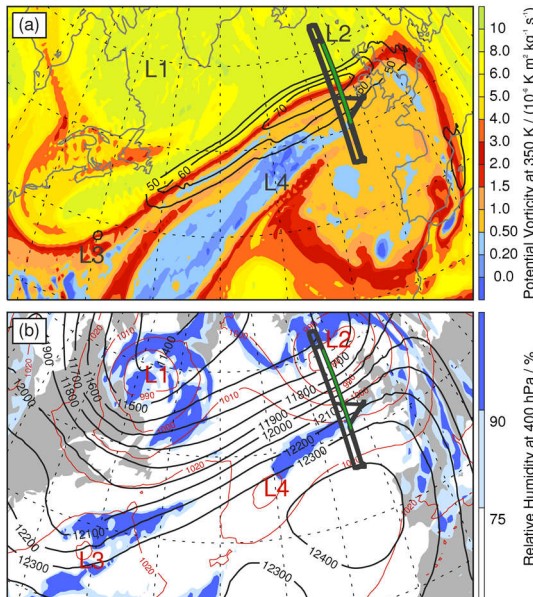

**Figure 1:** (a) Potential vorticity (in colours, $10^{-6}$ K m² kg⁻¹ s⁻¹ = 1 PVU) and horizontal wind speed (black contour lines, in m s⁻¹ for > 50 m

s⁻¹) at 350 K. (b) Relative humidity at 400 hPa (colours, in %), geopotential height at 150 hPa (black contour lines, in m) and surface pressure (red contour lines, in hPa) as represented in the ECMWF operational analysis at 1 October 2017, 18 UTC. L1–L4 mark the location of surface cyclones. (a) and (b) are superimposed by the HALO flight track (12:05–21:57 UTC, thick black line) and the subsection from 18:40 to 20:00 UTC (green line) that is discussed in this paper.



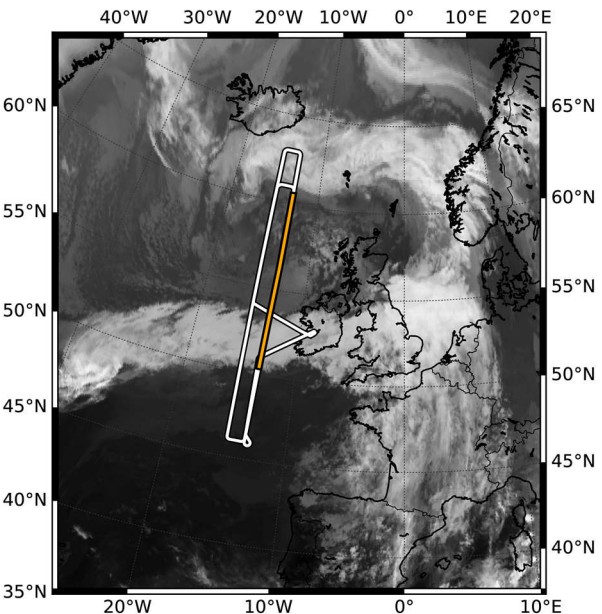

**Figure 2:** Meteosat SEVIRI infrared satellite image (10.8 μm) for 1 October 2017, 18 UTC superimposed by the HALO flight track
(12:05–21:57 UTC, thick black line) and the subsection from 18:40 to 20:00 UTC (orange line). Copyright 2020 EUMETSAT.

### 3.2 O$_3$ and H$_2$O along the observed cross-sections

Figure 3 shows the observed distributions of H$_2$O and O$_3$ along the above described ~1100 km long meridionally oriented
cross-section between 18:40 and 20:00 UTC comprising a total of 200 lidar profiles that are shown for an altitude range
between 3.5 and 14 km. In the first 400 km, the dynamical tropopause (2 PVU) is located at altitudes between 7.5 and 8.5 km
before it slopes down to about 5 km altitude. At about half of the flight leg, a steep ascent of the dynamical tropopause is
accompanied by tropospheric air masses reaching altitudes up to ~14 km further south. The stratospheric air is characterized
by increased static stability as visible from the increased vertical potential temperature gradient. Westerly jet stream winds
with maximum wind speeds up to 90 m s$^{-1}$ at 9 km altitude and strong horizontal and vertical wind speed shear blow
perpendicular to the flight track. Isentropes intersect the dynamical tropopause between 314 and 366 K of which the lowest
ones extend downward in association with folding of the tropopause which is typically initiated by ageostrophic circulation
around the jet stream causing an intrusion of stratospheric air (Keyser and Shapiro, 1987).

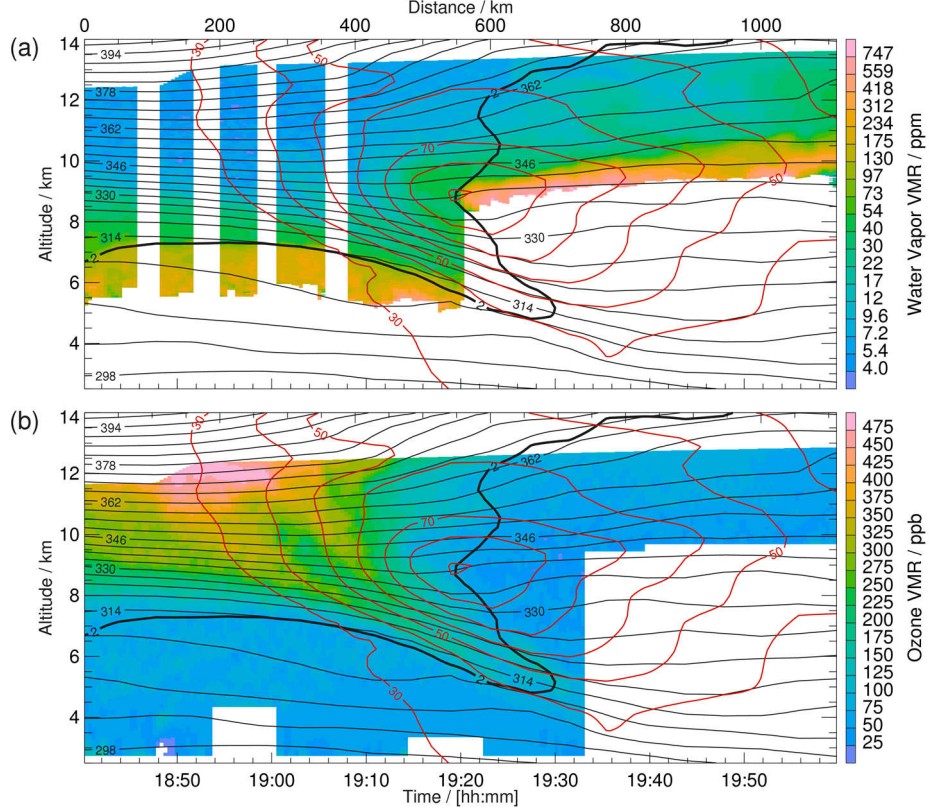

**Figure 3**: DIAL observations (in colours) of (a) $H_2O$ volume mixing ratio (VMR, in ppm = $10^{-6}$ = µmol mol$^{-1}$) and (b) $O_3$ VMR (in ppb = $10^{-9}$ = nmol mol$^{-1}$) on 1 October 2017 (see Fig. 1 for the flight track). (a) and (b) are superimposed by horizontal wind speed (red contours, in m/s for wind speeds > 30 m s$^{-1}$) potential temperature (black contours, in K) and dynamical tropopause (2 PVU, thick black contour) interpolated from one-hourly ECMWF ERA-5 reanalyses.

$H_2O$ shows the lowest VMRs (3–7 ppm) at the highest potential temperature in the stratosphere which are typical values for autumn in the lowermost stratosphere (e.g. Zahn et al., 2014). In contrast, the highest $H_2O$ VMRs occur in the troposphere to the north and south of the jet stream ranging from ~100 to 1000 ppm. The moist tropospheric air north of the jet stream is relatively well-mixed and provides typical autumnal values for the upper troposphere of 100–300 ppm. The high-reaching tropospheric air to the south of the jet stream is much more stratified. $H_2O$ quickly decreases above a shallow (1–1.5 km) moist layer (100–1000 ppm) in the lowest part. The highest $H_2O$ VMRs are indicative for recent vertical transport (e.g., in WCBs) from the moist lower troposphere (Zahn et al., 2014). Furthermore, the stratification and some filamentary structures with enhanced $H_2O$ at upper levels suggest differential transport impacting the distribution of the upper-tropospheric air. The





lowest $H_2O$ VMR (10–40 ppm) at the highest levels and the exceptionally high dynamical tropopause and potential temperatures are indicative for transport from the subtropical or tropical UT that was shown for individual case studies in comparable dynamical situations (Pan et al., 2007; Vogel et al., 2011) and in climatological distributions (Hegglin et al., 2009; Zahn et al., 2014). Missing data below the moist layer stems from mid-level clouds (see Fig. 1 and 2) while, in the first

half of the flight, the data gap below ~5.5 km results from reduced energy and attenuation of the laser signal in lower and moister air. Vertical stripes of missing $H_2O$ data are the result of cooling issues intermittently occurring at high flight altitudes with high potential temperatures.

In addition, observed $O_3$ values represent typical concentrations for the season (c.f. Krebsbach et al., 2006). In contrast to $H_2O$, highest $O_3$ ($O_3$ VMR of 300–500 ppb) was measured in the lowermost stratosphere (LMS) with a strong decrease

towards lower altitudes and the south. Although the region of highest $O_3$ is relatively compact, it shows large inhomogeneity on smaller scales with two filamentary structures of increased VMR extending across isentropes towards the intrusion located below the jet stream where air is adiabatically transported towards the ground. In the troposphere, $O_3$ is comparatively low (20–100 ppb) with the lowest values occurring in the mid-troposphere to the south of the jet stream being indicative for recent transport from the lower troposphere. The tropopause fold redistributes $O_3$ and $H_2O$ with $O_3$ decreasing

and $H_2O$ increasing at its sides.

Note that only collocated data along the cross-section is used in the following T-T diagnostics which covered the lower stratosphere north of the jet stream and a part of the upper troposphere on both sides. Therefore, it is well suited to investigate the ExTL. Note that due to the presence of clouds no collocated observations are available in the lower part of the tropopause fold.

The characteristic opposite behaviour of $O_3$ and $H_2O$ in the ExTL is particularly visible in Fig. 4 showing individual profiles in the northern part of the flight before 19:18 UTC. Profiles are shown in tropopause-relative coordinates (e.g. Birner, 2002) which pronounces the tropopause-based trace gas gradients (Hoor et al., 2004) as the dynamical tropopause slightly decreases southward. Between 18:50 and 19:08 UTC (bluish profiles) tropospheric $H_2O$ and $O_3$ VMRs are relatively constant with height and range between 100 and 300 ppm for $H_2O$ and 50 and 100 ppb for $O_3$. $H_2O$ VMRs decline above the

tropopause to LMS background values (3–7 ppm) at 3 km above the tropopause. $O_3$ instead steadily increases above the dynamical tropopause. It is the 2 km thick transition region above the tropopause that provides intermediate characteristics between stratosphere and tropospheric values of $O_3$ and $H_2O$ which is indicative for mixed air masses in the ExTL. When approaching the jet stream (greenish profiles, 19:08 to 19:18 UTC) higher $H_2O$ VMR with increased variability and corresponding lower $O_3$ VMR suggest the influence of mixing processes in this altitude range.

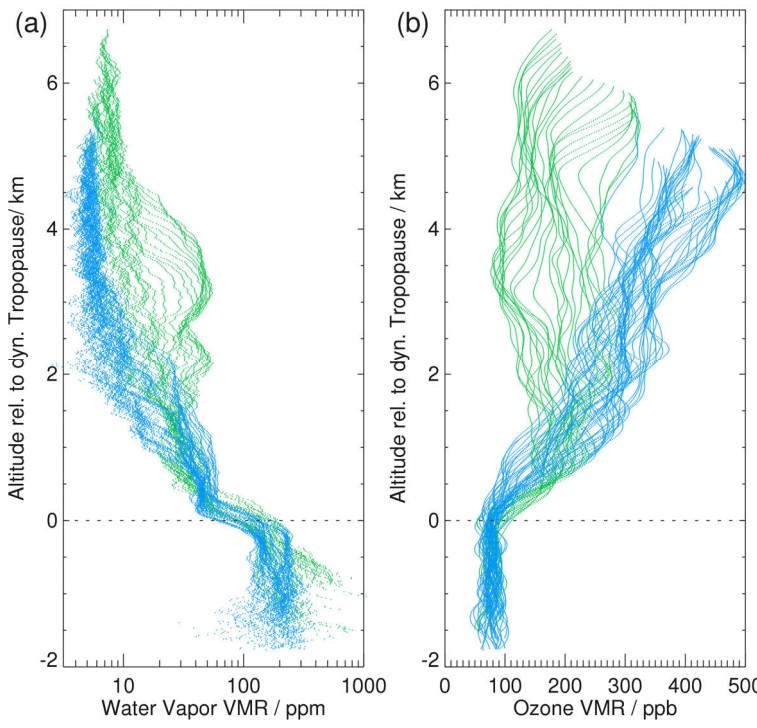


**Figure 4**: Profiles of (a) $H_2O$ and (b) $O_3$ VMR relative to the dynamical tropopause (2 PVU). Blue dots show profiles for the time period 18:40 to 19:08 UTC and green dots for 19:08 to 19:18 UTC for collocated data along the cross-section in Fig. 3.

### 3.3 $O_3$ and $H_2O$ as tracer-tracer correlations

In T-T space, the collocated $O_3$ and $H_2O$ lidar measurements (Fig. 5a) form an L-shaped distribution with an arc-shaped

transition in-between which immediately highlights that the DIAL observations are suited to distinguish stratospheric, tropospheric and mixed air masses. Note that the non-linear shape of the mixing lines results from the logarithmic scale of $H_2O$, which was also observed in other studies using $H_2O$–$O_3$ correlations (Hegglin et al., 2009). In order to better characterize the partly superposed measurements, Fig. 5b shows the number of observations contributing to individual bins in the T-T diagram. $O_3$ values of less than ~100 ppb allow a first rough depiction of the *tropospheric branch* that holds

variable $H_2O$ with VMRs covering four orders of magnitude and featuring two maxima between 10 and 40 ppm and 100 and 300 ppm. The *stratospheric branch* is immediately identified by $H_2O$ observations with VMR of less than ~7 ppm. In between both branches, collections of mixing lines form compact traces with increased numbers of observations that will be called mixing regimes in the following. The arc-shaped mixing regime, that is split in two traces at higher ozone





concentrations, connects stratospheric $O_3$VMRs of 250–350 ppb with tropospheric $H_2O$ VMRs of 30-70 ppm. Interestingly,
an area to the right of this arc-shaped mixing regime with a low number of observations suggests mixing between already
mixed air and more humid ($H_2O$ VMR of 100–200 ppm) tropospheric air. The lower left area in T-T space, connecting very
dry tropospheric air ($H_2O$ VMR of 8–15 ppm) and ozone-rich stratospheric air ($O_3$VMR > 250 ppb) is less obvious as it may
be part of the dry and ozone-poor stratospheric branch typically originating from low-latitudes (e.g. Tilmes et al., 2010) or it
may be related to mixing of dry subtropical tropospheric air with ozone-rich stratospheric air. The former appears less likely,
as climatological mid-latitude distributions show such pure stratospheric air masses at very low $H_2O$ only during northern
hemispheric winter (Hegglin et al., 2009). Additionally, $H_2O$ is slightly increased compared to the stratospheric background
with $H_2O$ VMR of 4–7 ppm (c.f. Zahn et al., 2014) leading to a somewhat reduced slope compared to the stratospheric
branch. Please note that the above diagnosed tropospheric and stratospheric entry VMRs represent typical values with
respect to climatology (Hegglin et al., 2009).Based on these findings, Fig. 5c introduces a classification of the observed air
masses solely based on their location in T-T space for this instantaneous jet stream cross-section. The light blue area covers
the stratospheric branch (STRA) with VMR $O_3$ > 280 ppb and VMR $H_2O$ < 7 ppm. The tropospheric branch is subdivided
into three classes with slightly varying $O_3$ thresholds depending on the measurement frequencies in Fig. 5b. TRO-1
represents the driest tropospheric air (VMR $H_2O$ < 30 ppm), TRO-2 the intermediate air mass (30 ppm < VMR $H_2O$ > 100
ppm) and TRO-3 the moistest air mass (VMR $H_2O$ > 100 ppm). Table 1 summarizes the applied thresholds to detect
tropospheric and stratospheric air. The three above discussed mixing regimes are coloured in dark green (MIX-1), green
(MIX-2) and light green (MIX-3) and connect the stratospheric branch with different parts of the troposphere. The threshold
between TRO-1 and TRO-2 was adapted to represent the tropospheric end of transitions within MIX-1 and MIX-2. The
distribution of these air mass classes in geometrical space are further detailed in Sect. 3.4.

**Table 1:** Thresholds used for air mass classification of tropospheric and stratospheric air in T-T space (see also Fig. 5).

| Class | $O_3$ / ppb | $H_2O$ /ppm |
|---|---|---|
| STRA | > 280 | < 7 |
| TRO-1 | < 90 | 10 - 30 |
| TRO-2 | < 80 | 30 - 100 |
| TRO-3 | < 100 | > 100 |






Figure 6 shows mean and variability of pressure, potential temperature and PV for each bin in the T-T diagram. STRA provides low pressures and high potential temperatures that decrease towards lower $O_3$ which corresponds to vertically decreasing $O_3$ values (Fig. 3b). The increased variability in pressure at lower $O_3$ VMRs within STRA can be explained by the
additional latitudinal decrease of $O_3$ at the highest altitudes (low potential temperatures). The dry tropospheric air mass TRO-1 also provides low pressure and high potential temperature. Conversely, TRO-2 and TRO-3 possess higher pressure and lower potential temperature. The lowest tropospheric $O_3$ corresponds to tropospheric and ozone-poor air at ~8–10 km on the southern side of the jet stream. Towards higher $O_3$, pressure increases (potential temperature decreases) within TRO-2 and TRO-3 accompanied by increased variability which corresponds to observations at different altitudes on the northern and
southern side of the jet stream. MIX-1 features low pressures connecting TRO-1 and STRA indicating that the transition occurred at high levels. High and relatively constant potential temperatures suggest an isentropic mixing regime within MIX-1. The lower the pressure (the higher the potential temperature) within MIX-1 the lower the $H_2O$ concentration was which is expectable due to vertically decreasing tropopause temperatures that define the ExTL moisture when mixed across the jet stream (e.g. Zahn et al., 2014). Within MIX-2, pressure and its variability increase towards the tropospheric branch while
potential temperature decreases. The higher the altitude (higher potential temperature and lower pressure), the lower the $H_2O$ VMR and $O_3$ VMR were observed in MIX-2. The area within MIX-2 that shows as separated and rather linear trace (Fig. 5b) features highest potential temperature and lowest pressure. MIX-3 occurs at low potential temperatures and higher pressures. Figure 6c shows the distribution of PV which is a tracer for stratospheric air comparable to $O_3$ as it has comparable gradients across the tropopause with high values in the stably stratified stratosphere (e.g. Shapiro, 1980). PV is conserved in adiabatic
motions. However, diabatic processes such as cloud condensation, radiation or turbulent mixing impact the PV distribution (Spreitzer et al., 2019). As expected, PV is highest in STRA and strongly decreases along the mixing regimes with decreasing $O_3$. The mixed air masses show PV values between 2 and 7 PVU. Furthermore, PV is found to be variable within the regimes in the ExTL. Therefore, it is assumed that the interrelation of the dynamical and chemical transition depends on altitude (potential temperature), a circumstance that will be discussed in more detail in Sect. 3.5. Lowest PV values are
correlated with low $O_3$ values in the troposphere indicating more recent transport from the lower troposphere.





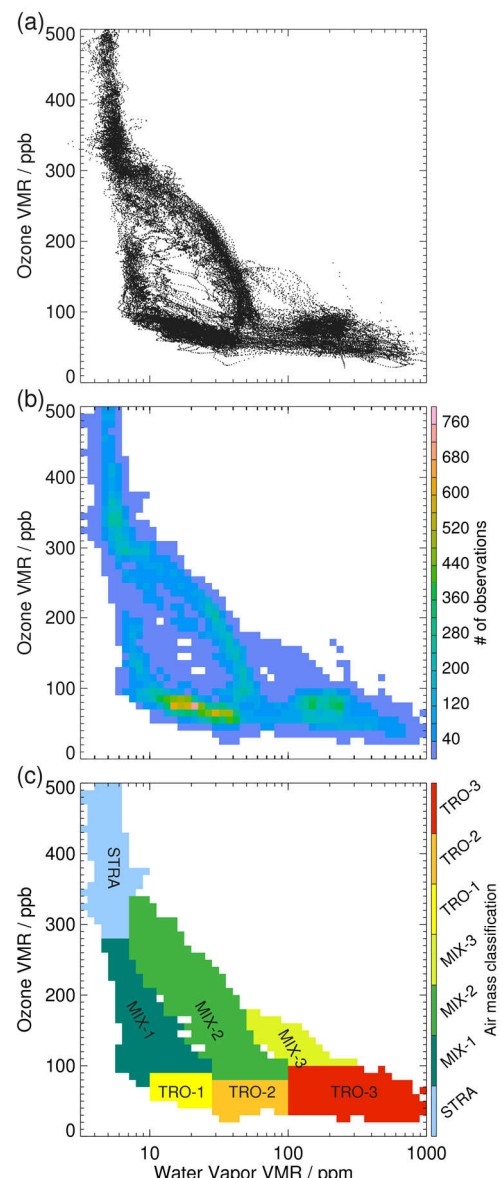

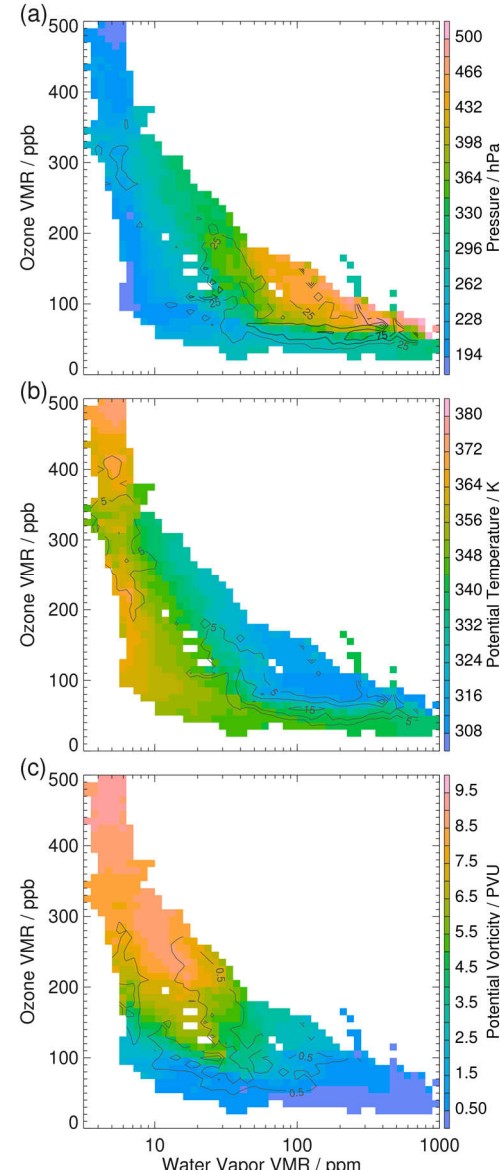

**Figure 5:** Tracer-tracer (T-T) correlations of $O_3$ and $H_2O$ for the collocated DIAL data on 1 October 2017. (a) All collocated observations. (b) Number of data points and (c) air mass classification for bins in T-T space (bin sizes of 10 ppb for VMR $O_3$ and 0.05 of $\log_{10}$(VMR $H_2O$/ppm)).

**Figure 6:** T-T correlation as in Fig. 5c but for mean (in colours) and standard deviation (grey contours) of (a) pressure, (b) potential temperature and (c) potential vorticity.



## 3.4 A combined view of O₃ and H₂O in T-T and geometrical space

A back projection of the air mass classification from T-T space (Fig. 5c) into geometrical space along the cross-section gives more detailed information on the shape and composition of the ExTL (Fig. 7a). First, it is striking that the different air masses correspond to remarkably coherent areas along the cross-section. The mixed air masses in the ExTL (MIX-1, MIX-2

and MIX-3) reach about ~30 K above the tropopause in the northern part before they ascend with the rising tropopause further to the south. MIX-1 occurs at highest altitudes in the upper part of the jet stream with its bottom being relatively well defined by the 348 K isentrope. MIX-1 connects STRA with TRO-1 (isentropic transition) which underlines the validity of considering MIX-1 as being mixed air instead of stratospheric background. The relatively constant potential temperature along the mixing regime (Fig. 6b) suggests the relevance of mixing processes in the upper-part of the jet stream, i.e. a region

known for turbulence-induced exchange between neighbouring air masses (Shapiro, 1980).

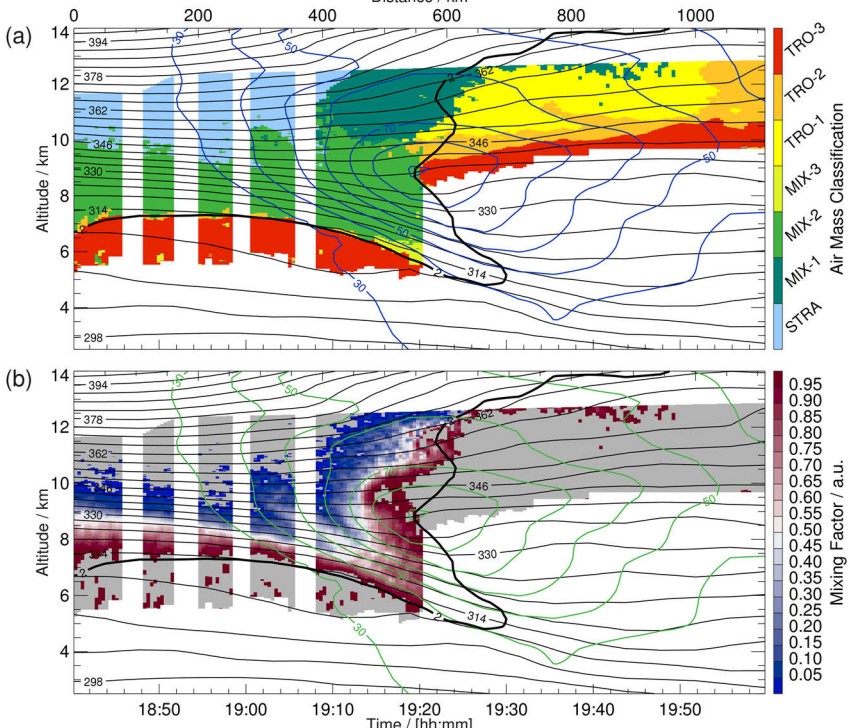

**Figure 7:** (a) Collocated observations coloured by (a) the air mass classification in Fig. 5 and (b) the mixing factor diagnostic as shown in Fig. 8a (for details see text).


Below, MIX-2 comprises isentropic transitions of STRA with TRO-2 and TRO-3 above the clouds in the troposphere as well as cross-isentropic vertical transitions in the northern part of the flight between STRA and TRO-3. In the northern part of the flight section, the bottom of the ExTL (MIX-2) agrees with the dynamical tropopause which confirms findings by Pan et al. (2007). Near the jet stream the agreement of the almost vertical dynamical tropopause and the border to the tropospheric air (TRO 1–3) is not as uniform and TRO-1 and 2 air masses reach into areas north of the dynamical tropopause. MIX-3 occurs in the lowest observed part of the intrusion that was observed before reaching the clouds. The geometrical location points to mixing processes between mixed ExTL air masses and the TRO-3 air mass below in the lower part of the jet stream. Note that the 348 K isentrope also separates TRO-1 from TRO-2 in the tropospheric air, which points to different source regions of humidity in the troposphere. Stratospheric air (STRA) is located at the highest altitudes and in the most northern part.

To further investigate the structure and strength of stratosphere to troposphere transitions within the ExTL, a mixing degree metric is adapted to the lidar data which was initially introduced for in situ observations by Kunz et al. (2009). In that publication, the mixing degree is a measure of how much mixed air masses deviate from the background due to mixing in their history. It is solely based on the location of the observed air mass in T-T space and increases with distance from the undisturbed stratospheric and tropospheric background and with distance from the intersection point of the two branches. Here, a comparable mixing factor is determined, however, unlike Kunz et al. (2009), it does not account for the distance to the intersection point of tropospheric and stratospheric branches as a mixing event is not considered to be stronger in case the tropospheric $H_2O$ VMR and stratospheric $O_3$ VMR are increased. At first dimensionless variables x and y are calculated from the VMR $H_2O$ and VMR $O_3$ with x = $(\log(H_2O) - \log(H_2O_{MIN}))$ /$(\log(H_2O_{MAX}) - \log(H_2O_{MIN}))$ and y = $(O_3 - O_{3,MIN})/(O_{3,MAX} - O_{3,MIN})$ using the thresholds $H_2O_{MIN}$ = 6.5 ppm, $H_2O_{MAX}$ = 1000 ppm, $O_{3,MIN}$ = 90 ppb and $O_{3,MAX}$ = 500 ppb to select the mixed air mass. The observations with lower VMR than this range are considered to be unmixed tropospheric and stratospheric air. In order to range from 0 (pure stratospheric air) to 1 (pure tropospheric air) the mixing factor f is calculated for x > y as f = 1 - (0.5*(y/x)) and for y < x as f = 0.5*(x/y). Figure 8a shows the observations color-coded by the mixing factor. Please note that unlike the earlier presented air mass classification in T-T space, the thresholds are defined constant but the mixed air masses approximately correspond to each other in geometrical space (c.f. Fig. 7). Within the ExTL the mixing factor picks up the major transition regions MIX-1, MIX-2 and MIX-3. At the highest levels (above 350 K,), i.e. directly above of the jet stream maximum winds, the isentropic transition is rather uniform compared to the layer beneath (340–350 K). Below potential temperatures of 340 K, the transition is again more uniform. Within the tropopause fold air masses with intermediate chemical composition are transported towards the lower troposphere. The





above described stratospheric filaments of high O$_3$ correspond to decreased mixing factors indicating stratospheric character of the observed air compared to the surrounding.

When the mixing factor is presented with respect to potential temperature (Fig. 8b) three different characteristic transitions appear. *First*, cross-isentropic transitions between 310 and 340 K correspond to the vertical transitions in the first half of the flight i.e. transitions from TRO-3 via MIX-2 into STRA (see Fig. 5 and Fig. 7). *Second*, isentropic transitions appear between 350 and 365 K that correspond to high level isentropic transitions between STRA and TRO-1 via MIX-1. Rapid transitions at and above the level of the jet stream result in a lack of observations with intermediate chemical characteristics for potential temperature between 340 and 350 K. *Third*, transitions from mixed air to tropospheric air with increasing potential temperatures (325–340 K) are related to transitions in the upper part of the tropopause fold (see Fig. 7b). Figure 7 and 8 highlight the distribution of air masses as classified in T-T space and suggested that both vertical and isentropic transitions did occur. MIX-1 is characterized by isentropic transitions while MIX-2 is also influenced by vertical transitions.

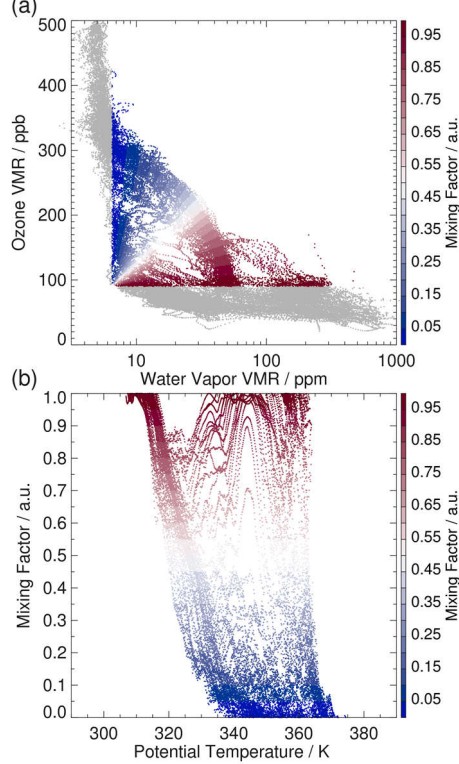

**Figure 8:** Mixing factor diagnostic for all collocated measurements in (a) T-T space using lower limits for H$_2$O VMR of 6.5 ppm and for O$_3$ VMR of 90 ppb for selecting mixed air masses (for details see text) and in (b) as a function of potential temperature and mixing factor.





Figure 9 shows how observations in certain subregions along the lidar cross-section become apparent in T-T distributions which only becomes possible through the application of the novel collocated lidar data. Profiles before 19:00 UTC (Fig. 9a) represent vertical transitions in the first part of the flight that form transitions from high $O_3$ in the stratosphere along the arc-shaped mixed region in T-T space into TRO-3 (Fig. 9b). Within the ExTL (MIX-2) the mixing lines connect comparably

high stratospheric $O_3$ (~300 ppb) with high tropospheric $H_2O$ (~50 ppm). The tropospheric air () is rather rich in $O_3$ and does not reach the highest levels of $H_2O$ within TRO-3. Fig. 9c and d represent the layer 335 to 340 K between 19:00 and 19:45 UTC representing the above discussed rapid transition between tropospheric and ExTL air at the level of the jet stream maximum connecting TRO-3 and MIX-2. Both air masses are clearly separated in T-T space with $H_2O$ VMRs jumping from ~50 ppm at the tropospheric end of MIX-2 to ~500 ppm in TRO-3 (Fig. 9b) over a very short spatial distance (Fig. 9c)

indicating minor mixing between both air masses. The more stratospheric part of MIX-2 follows the arc-shaped distribution while, the tropospheric part that is facing the jet stream is slightly detached (Fig. 9d), which potentially indicates different mixing processes within this particular layer. Interestingly, the layer directly above (340–347 K, Fig. 9e) that represents the transition of MIX-2 air with medium moist air in TRO-2 is still relatively abrupt but the discontinuity occurs within the ExTL (Fig. 9f), i.e. between the linear-shaped stratospheric part and the tropospheric end, suggesting some in-mixing of

TRO-2 air into this layer. The location of these abrupt transitions in the two layers between 337 K and 347 K occur at the same spatial location (c.f. Fig. 7b). Both layers do not reach far into the LMS. Note that due to mixing along its Lagrangian history it may be that the chemical composition is influenced by earlier events or processes, which explains the complexity of the distributions in T-T space. However, as both layers show rapid jumps in T-T space between neighboring air masses, recent mixing is expected to be rather weak. This agrees with the concept of turbulent mixing being more intense in the

upper and lower part of the jet stream (Shapiro, 1976). Indeed, the situation appears different in the upper part between 349 K and 358 K where the mixing factor diagnostic shows more uniform transitions (Fig. 7b). The layer that connects LMS air in STRA with low $H_2O$ VMRs in TRO-1 (Fig. 9g) shows much more gradual transitions across MIX-1 in T-T space (Fig. 9h). Please note that the minimum of observations within MIX-1 ($O_3$ VMR 200–250 ppm) is a result of the data gap between 19:06 UTC and 19:08 UTC. Figure 9i and j show the vertical transitions at the bottom of the tropopause fold that unlike the

vertical transitions in Fig. 9a, b show a mixing of ExTL air (MIX-2) and tropospheric air (TRO-3) below. This was most likely caused by recent turbulent mixing related to strong vertical shear in the lower part of the jet stream.



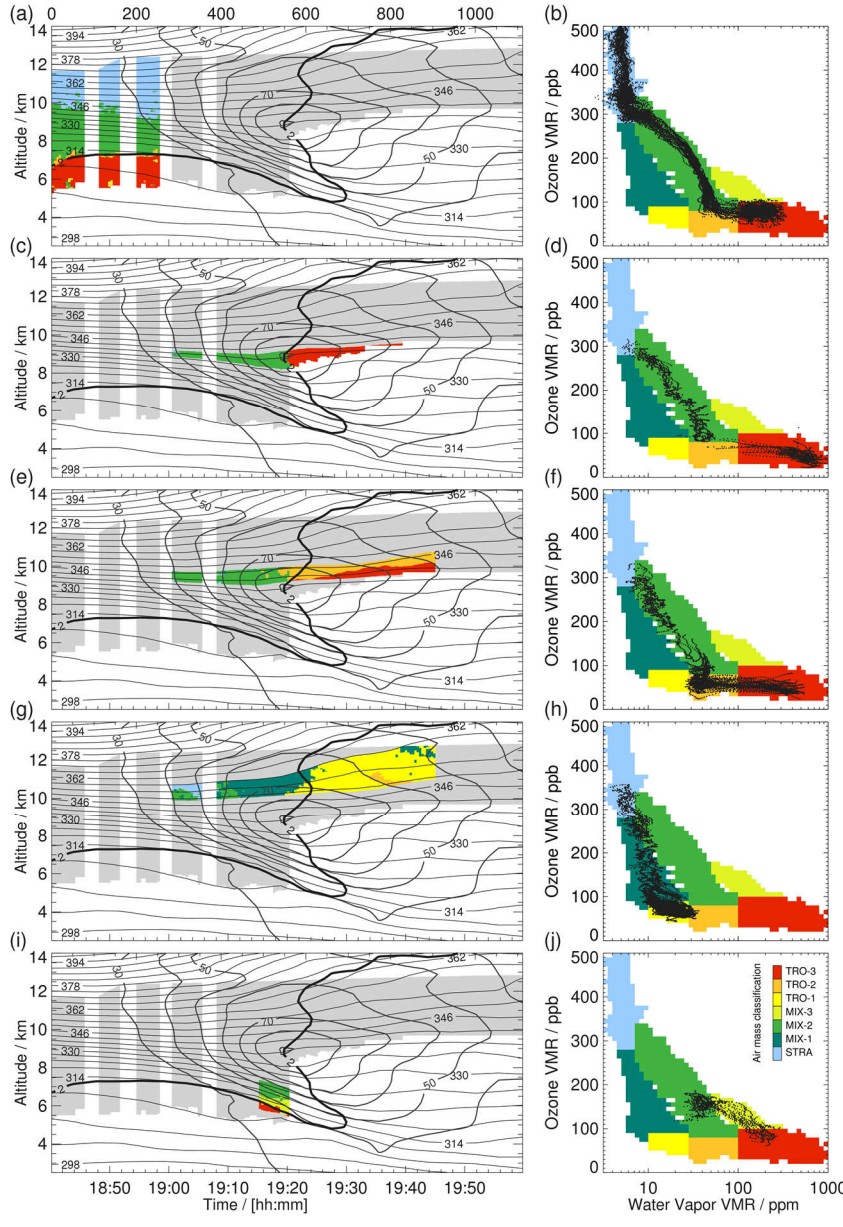

**Figure 9**: Data subsets of the collocated lidar data. (a, b) before 19:00 UTC with (a) showing the data as classified in Fig. 7a and in (b) as black dots superimposed on the classification of all data shown in Fig. 5c. (c, d) as (a, b) but for data in the time period from 19:00–19:45 UTC in the layer 335–340 K. (e, f) for 19:00 to 19:45 UTC and 340–347 K. (g, h) for 19:00 to 19:45 UTC and 349–358 K. (i, j) for 19:15 to 19:25 UTC and 311–324 K.




### 3.5 Isentropic trace gas gradients

The interrelation of chemical and dynamical discontinuities at the tropopause is of central interest to understand trace gas distributions and their relation to transport and mixing processes. The dynamical discontinuity is most often defined by the

dynamical tropopause, which runs vertically near jet streams as shown in the foregoing discussion, and is therefore suitable to characterize the ExTL along isentropic surfaces crossing it. However, the selection of a PV threshold for the dynamical tropopause is to some extent arbitrary and influences statistical analyses of the ExTL (Pan et al., 2004). In dynamic meteorology often the 2-PVU definition is applied while UTLS studies aimed to search the best agreement of the dynamical with the thermal tropopause (e.g. 3.5 PVU) which varies both regionally and seasonally (Hoerling et al., 1993). Sections 3.3

and 3.4 confirmed earlier findings that the dynamical tropopause (using a 2 PVU definition) marks the bottom of the ExTL in the situation of a flat tropopause. However, where the dynamical tropopause is almost vertical, its location relative to the ExTL boundary is found to be variable (Fig. 7a). Additionally, the shown mixing factor metric (Fig. 7b) indicates differing strengths in the ExTL transition at different layers within the jet stream.

To account for the ambiguous tropopause definitions, Kunz et al. (2011a) introduced a different definition of the dynamical

tropopause which is defined as the maximum isentropic PV-gradient which is maximized near jet streams (Martius et al., 2010). They constrained their PV-gradient-tropopause by the wind speed to correctly identify high wind speed situations near polar and subtropical jet streams. Besides a seasonal increase they found an average increase of PV at their PV-gradient-tropopause with increasing potential temperature, varying between 1.5 and 5 PVU between 310 and 350 K. In a second step, Kunz et al. (2011b) found better consistency of their PV-gradient-tropopause with isentropic trace gas gradients

of $O_3$ and CO compared to a fixed PV contour. In their study they used in situ CO and $O_3$ data from multiple research flights covering a period of three months and one-year data from a chemistry climate model to study the seasonal behavior of the chemical and dynamical discontinuities. For a case study they showed that the alignment of dynamical and chemical discontinuities holds for an instantaneous latitudinal cross-section using model data only. In the following we extend their analysis for our case using the lidar trace gas distributions to derive isentropic gradients and compare them with the various

tropopause definitions.

Figure 10a and b show regridded versions of the $O_3$ and $H_2O$ cross-sections using potential temperature as vertical coordinate. As in Fig. 3a and b, all available $O_3$ and $H_2O$ observations are used for maximum data coverage to determine isentropic gradients. Obviously, the stratospheric air with its strong vertical gradients in potential temperature gets stretched while the tropospheric part shrinks in isentropic coordinates. The jet stream maximum is located at approx. 340 K with the 2

PVU isoline crossing it. Higher PV isolines appear north of the jet. Following Kunz et al. (2011a), the PV-gradient-


tropopause is calculated along the isentropes in the latitudinal cross-section (see white dots in Figs. 10c and d). Note that the quasi-linear interpolation between gridded model data along the 15° W meridian resulted in angular PV distributions on isentropes that required smoothing of the isentropic PV using a moving average. For this reason, PV contours in Figs. 10c and d appear slightly smoothed. Consistent with Kunz et al. (2011a), the PV-gradient-tropopause is shifted to higher PV

values with increasing potential temperature. It approximately follows the curvy shape of individual PV isolines. Above and below the level of maximum winds in the region of maximum PV gradients, the PV-gradient-tropopause is shifted towards higher PV values.

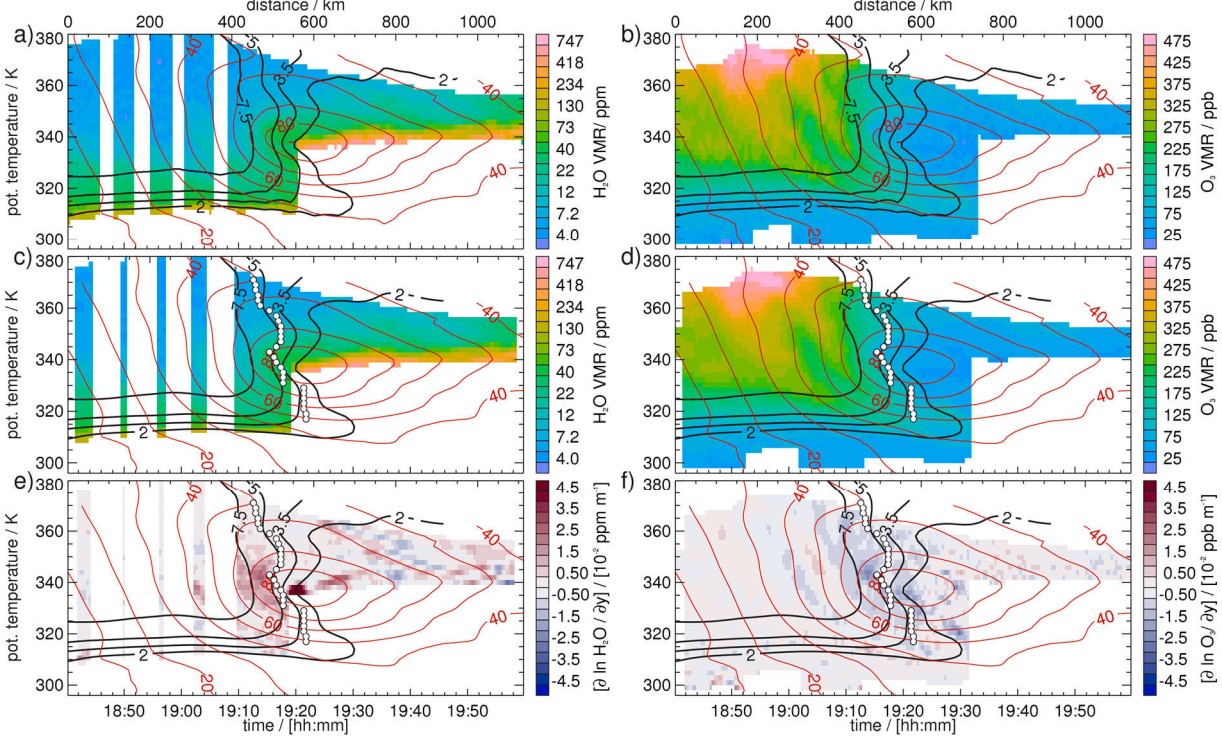

**Figure 10:** Regridded DIAL observations of (a) H₂O and (b) O₃ (as shown in Fig. 3) using potential temperature as vertical coordinate
(same profile locations and for potential temperature bins of 2 K). (c, d) as (a, b) but using a moving average filter along isentropic levels (using 7 observations for H₂O and O₃ and 13 model values for PV and wind speed). (e, f) show isentropic gradients of the natural logarithm of VMR H₂O and VMR O₃ based on the isentropically smoothed data. All panels are superimposed by PV contours (2, 3.5, 5 and 7 PVU, thick black contours) and wind speed (in m s$^{-1}$ for > 30 m s$^{-1}$). White dots in c–f mark the PV-gradient-tropopause based on maximum gradients of isentropic PV and winds following Kunz et al. (2011a) (for details see text).


To calculate isentropic trace gas gradients from the observational data, $O_3$ and $H_2O$ were also smoothed (Fig. 10c, d) to account for instrument-generated noise causing strong local gradients. Note that smoothing of the $H_2O$ data increased the data gaps before 19:10 UTC. Trace gas gradients were then calculated for the natural logarithm of the trace gas concentration as suggested by Kunz et al. (2011b) with the purpose of scaling down increased gradients in the source regions at higher concentrations of the trace gas (stratosphere for $O_3$ and troposphere for $H_2O$). This shifts the focus towards the gradients

across the tropopause. Strongest isentropic gradients of $H_2O$ are found at the level of maximum winds within the ExTL (MIX-2), while above $H_2O$ gradients are weaker, where tropospheric $H_2O$ VMR (TRO-1) is lower and mixing within MIX-2 is more uniform (Fig. 10e, f). Please note that the highest local gradient ~19:20 UTC is related to very high $H_2O$ VMR at the edge of the cloud layer. $O_3$ with higher coverage in the lower part of the tropopause fold also indicates maximum gradients at the level of maximum winds. Additionally, an increased gradient occurs at the bottom of the fold as compared to $H_2O$, $O_3$

gradients are smaller above 350 K. Some increased positive and negative gradients are found in the stratosphere related to the $O_3$ filament. The PV-gradient tropopause follows the regions of highest isentropic trace gas gradients much better than the 2-PVU isoline, especially in case of the $O_3$ gradients, which confirms Kunz et al (2011b). Kunz et al. (2011b) argue that the better agreement of the PV-gradient tropopause with the stratospheric tracer originates from their stratospheric concept, in the sense that the chemical tracer $O_3$ and the dynamical tracer PV are higher in the stratosphere. In contrast $H_2O$ exhibits

stronger tropospheric gradients which are mostly related to transport processes into the UT.

**4. Summary and Discussion**

In this study we analysed mixing of air masses at the extratropical tropopause that shapes the structure and the chemical composition of the ExTL, i.e. a region of central importance for weather and climate. For this purpose, we applied the first-ever set of collocated $O_3$ and $H_2O$ lidar observations obtained during the WISE field campaign over the North Atlantic

Ocean in autumn 2017. We demonstrate the potential of quasi-instantaneous cross-section observations across the tropopause to reveal the complexity in the two-dimensional $O_3$ and $H_2O$ distribution in a dynamically rather simple synoptic situation with a perpendicular crossing of a straight south-westerly jet stream. The presented flight on 1 October 2017 captured a low tropopause on the northern cyclonic shear side of the jet stream and a high tropopause with high-reaching tropospheric air to its south. In-between, a tropopause fold extended downward along tilted isentropes into the lower

tropospheric frontal zone before the tropopause strongly ascended accompanied by an upper-level frontal zone above the jet stream. This flight provides exceptionally good data coverage due to low cloud coverage beneath the aircraft and a high flight altitude. Established T-T diagnostics were applied to the lidar data to identify the ExTL and to depict its shape and



composition in new detail. Through a combination with a back projection of T-T-derived information to geometrical space, i.e. along the cross-section, a physically meaningful selection of thresholds needed for the air mass classification became

possible that is to some extent always arbitrary (e.g. Pan et al. 2004).

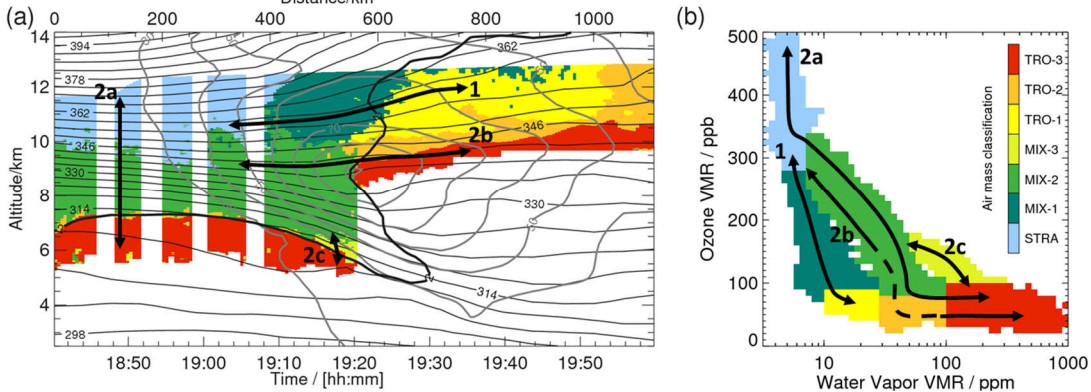

**Figure 11:** (a) as Fig. 7a and (b) as Fig. 5c but superimposed by the main ExTL transitions.

Up to now, collocated observations of tropospheric and stratospheric trace gases for individual dynamic situations were mostly restricted to a limited number of in situ flight legs at different altitudes and could only locate mixed, stratosphere and

tropospheric air masses along these legs projected onto cross-sections (e.g. Pan et al. 2007; Vogel et al., 2011; Konopka and Pan, 2012). Here, we show a first two-dimensional depiction of the extratropical tropopause that is summarized in a combined view of geometrical and T-T space in Fig. 11 showing the major transitions between stratosphere and troposphere. Using probability densities in T-T space allowed to identify certain clusters of mixing lines (mixing regimes) and subdividing mixed (MIX-1, MIX-2, MIX-3) and tropospheric air (TRO-1, TRO-2, TRO-3). In the upper part of the jet

stream, above 348 K, isentropic transitions (transition 1 in Fig. 11) connect ozone-rich stratospheric air (STRA) and dry tropospheric air (TRO-1) via mixing regime MIX-1. The second major transition (2a and 2b in Fig. 11) involves mixing regime MIX-2 and connects stratospheric air (STRA) with moister tropospheric air (TRO-2 and TRO-3). Both, cross-isentropic (transition 2a in Fig. 11) and isentropic transitions (2b) contribute to the distribution of MIX-2 in T-T space. The isentropic transitions below 348 K at the level of maximum winds do not reach far into the stratospheric air and, most

strikingly, are separated in T-T space indicating reduced mixing. A mixing factor metric adapted from Kunz et al. (2009) confirmed the impression of little mixing at the level of the jet stream and more uniform transitions above. Additionally, the consideration of isentropic trace gas gradients showed maxima at the same levels. We also related the chemical to the dynamical discontinuity at the tropopause and could confirm that observed isentropic trace gas gradients follow a PV-





gradient-tropopause as proposed by Kunz et al. (2011b). In the lowest observed part of the fold, the transition regime MIX-3
(2c in Fig. 11) suggested mixing between ExTL and moist tropospheric air beneath, i.e. a region that is known for turbulence-induced mixing due to strong wind speed gradients (Danielsen et al. 1968; Esler et al. 2003; Cooper et al. 2004). Our findings confirm and visualize early concepts of turbulence induced mixing in strong wind shear regions above and below the maximum winds in the jet stream (Shapiro, 1976).

Several case studies documented transport pathways and mixing processes near the extratropical tropopause influenced by a
large dynamical variability. For the presented case we could show a clear separation of mixed air masses in T-T space (MIX-1 and MIX-2) although the isentropic transitions were stacked directly on top of each other. Due to the separation in the tropospheric and mixed air at the same potential temperature, we hypothesize that different transport pathways brought air with differing $H_2O$ VMRs to the upper-tropospheric end of the mixing regimes MIX-1 and MIX-2. Dry low latitude tropospheric air, being either dehydrated in the convective tropics or loosing moisture through lifting to a colder and higher
level in a tropical cyclone, arrived above moister extratropical air, typical for the warmer temperatures near the extratropical tropopause (e.g. Hegglin et al., 2009; Zahn et al., 2014). Potentially, Hurricane Maria, which was located in the tropical and subtropical western Atlantic in late September 2017 may have played a role (NOAA/NHC, 2018). A transport of low latitude tropospheric air was shown by Vogel et al. (2011) in the upper-troposphere on the anticyclonic shear side in a case study of a mid-latitude jet stream using in situ trace gas observations and Lagrangian model simulations. In this region featuring an
elevated tropopause, the model simulation indicated mixed air which could partly be confirmed by the observations. In our case no observations were available directly below the tropopause at ~14 km altitude. However, the southward extension of MIX-1 across the dynamical tropopause at highest level gives some indication for enhanced upper-level mixing. Pan et al. (2007) observed a comparable separation in T-T in a likewise synoptic setting of a mid-latitude jet stream crossing, however, the very low $H_2O$ with increasing $O_3$ were of stratospheric origin in this case observed in December.

The simultaneous contribution of extra-tropical and tropical tropospheric air to mixing in the ExTL contrasts with the conceptual model of distinct mixing processes at the subtropical and polar jet stream (e.g. Gettelman et al., 2011). Although the described case is special in terms of its transport, the distribution resembles the satellite-derived climatological T-T distribution of the northern hemispheric mid-latitudes (see Fig. 5 in Hegglin et al., 2009) that also shows mixed air masses with lower $H_2O$ beside the typical extratropical mixing lines. A clear separation of both regimes is not occurring in the more
uniform climatological data that is, however, averaging over a series of individual dynamic situations. Zahn et al. (2014) investigated processes and pathways controlling $H_2O$ in the UTLS and found an influence of tropical tropospheric air in



summer that they attributed to subsiding subtropical and tropical air masses in the downward branch of the Hadley cell. Thus, this case is considered as a representative set of observations of the ExTL for the season.

The interpretation of trace gases observations with opposing gradients across the tropopause in the context of T-T diagrams

is an established way to investigate ExTL (Gettelman et al., 2011). Mixed air mases showed up as compact regions in T-T space even if they were observed in various differing dynamical situations throughout a multi-week campaign period (e.g. Pan et al. 2004) or for multi-month averaged satellite data (e.g. Hegglin et al., 2009) which made the T-T method a valuable tool to determine the structure and composition of the ExTL. However, it raises the question whether mixing lines for individual case studies can be interpreted as causal physical links between neighboring air masses or whether the relatively

small variability of upper-tropospheric $H_2O$ and of lower stratospheric $O_3$ on such time scales is key to the compact distribution. The combined approach of geometrical and T-T space gave some new insight. We found both vertical and horizontal (isentropic) transitions between tropospheric and stratospheric background air forming mixing lines (see Fig. 11). It is difficult to imagine a process that physically links the tropospheric with stratospheric air up to 40 K higher in the first part of the flight. This suggests that the formation of the ExTL observed in this region is rather related to advection of mixed

air. In contrast, mixing in the upper and lower part of the jet stream is likely to have happened more recently due to shear-induced turbulent mixing. However, even when assuming stationarity of the flow over the past hours, the observed air masses will be separated over a short period of time due to the strong wind speed shear in the jet. The individual location in T-T space is rather an effect of mixing events in the Lagrangian history of the observed air than an effect of instantaneous mixing that may be suggested by a snapshot taken from the lidar. Although the combination of methods does not inform

about the process, the location and the time of the mixing event that has formed the individual ExTL observation, it generates additional value as it allows subdividing air masses and revealed certain sub-regimes of mixing that are most likely a signature of mixing on different timescales depending on the dynamical background.

Konopka and Pan (2012) showed that the ExTL formation is influenced by processes on synoptic time scales and highlighted that near the jet stream processes in the last 3 days were particularly important. In a follow-up study we aim to address the

question how transport has affected the distribution of trace gases for this particular case and what timescales did impact the different parts of the ExTL by adding a Lagrangian diagnostic. A combination of remote sensing data and in situ observations, e.g. of other trace gas species is envisaged to help to further evaluate these results. In the future such DIAL observations of $O_3$ and $H_2O$ may be applied to research efforts to investigate various other synoptic situations leading to mixing across the tropopause, e.g. above WCBs. Additionally, such two-dimensional observations of $O_3$ and $H_2O$ may be of

high relevance for the validation of chemistry and numerical weather prediction models, which suffer from a lack of

operational data availability in the ExTL (Magnusson and Sandu, 2019) and rely on a realistic representation of mid-latitude dynamics, accurate parametrization of subgridscale mixing processes and realistic chemistry.

## Data availability

The lidar data used in this study is available through the HALO database (https://halo-db.pa.op.dlr.de/). We are grateful to
ECMWF granting access to the full-resolution ERA5 data. The ERA5 data have been downloaded and made available by Dr. Michael Sprenger from ETH Zurich. The ETH access to the ECMWF data is provided by the Swiss National Weather Surface (MeteoSwiss). Meteosat-10 L1 data (HRIT) were provided by EUMETSAT via EUMETCast.

## Author contribution

AS designed the study, performed the data analysis, produced the figures and wrote the text. MW performed the data
analysis of the DIAL water vapor and ozone data. AF performed DIAL observations during WISE. MW and AF advised on the analysis, contributed with ideas, helped with the interpretation of the data and commented on the paper.

## Competing interests

The authors declare that they have no conflicts of interest.

## Acknowledgment

The authors thank the WISE PIs and the flight planning team for supporting the remote sensing contribution to the campaign and for giving us insight in a new field of research. We are grateful to DLR for supporting this work in the framework of the DLR project "Klimarelevanz von atmosphärischen Spurengasen, Aerosolen und Wolken" (KliSAW). Additionally, we thank the German Science Foundation (DFG) for supporting the HALO contribution to the WISE campaign within the priority program SPP1294 HALO. We thank our colleagues Dr. Florian Ewald for creating the satellite image and Dr. Heidi
Huntrieser for her valuable comments to the manuscript.

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
