# Peer review of "Mixing at the extratropical tropopause as characterized by collocated airborne H2O and O3 lidar observations"

_Atmospheric Chemistry and Physics, 2020_

## Referee Comment (RC1) · Laura Pan (Referee) · 8 Dec 2020

Transport and mixing near the extratropical tropopause associated with the jet dynamics, relationship between the meteorological background and the chemical distribution in the extratropical upper troposphere and lower stratosphere (ExUTLS) have been an active line of research for multiple decades. An important development in the past two decades is the use of tracer-tracer correlation. This line of work has been largely fueled by new observations from new satellite and especially research aircraft data. The analysis presented in this work expands the horizon using the new generation of airborne lidar observations that provided co-located ozone and water vapor measure-

ments in a two-dimensional cross section intersects a region of active jet dynamics and stratospheric intrusion.

The authors did an excellent analysis to connect the 2D ozone and water vapor observation with all major prior studies when limited in situ sampling or coarse satellite data were available. The analysis combining the tracer space diagram and geometric space distribution using this unique 2D data is a significant contribution to the topic of ExUTLS research, and it could serve as a road map for analyzing future observations in this layer. Given that the modeling community is working toward improving spatial resolutions of the chemical transport and/or chemistry climate models to resolve the tropopause region, the method demonstrated here can be applied readily for process-specific model evaluations.

In summary, the new data presented is unique and of high value. The analysis presented contributes to the state of art on the topic of research. The large set of analysis details are presented in excellent clarity. I do have a few comments and suggestions below for the authors' consideration.

Major comments and suggestions:

1) It is too much of a missed opportunity not to include lapse rate tropopause in the analysis

An important question in the work of ExUTLS transport and STE is the relationship between various tropopause definitions, in particular, definitions of the dynamical tropopause and the temperature lapse rate based thermal tropopause from WMO. Chemical discontinuity and tracer-tracer relationship is a unique way to shed light on the physical behavior revealed by the relationship of the two definitions. The work did an excellent job diagnosing the dynamical tropopause using PV gradient versus fixed PV surface. It would be very valuable to diagnose the vertical discontinuity at the tropopause represented by the temperature lapse rate (discontinuity of the static stability) and PV contours, and the chemical mixing versus discontinuity associated with the

conditions where they agree or show large separation. In this direction, it is worthwhile to present the Lapse rate tropopause on the cross section, and to show additional LRT relative profiles in Fig. 4.

I understand that you don't have vertical legs for in situ measured temperature profiles but ERA5 300m resolution data should be able to provide a good approximation.

2) It is important to have a clear message that mixing processes that created these mixed regions did not happen in the observed cross section, but rather, upstream. As we draw connections between the air masses in the observed cross-section and identify the sources of mixing in the tracer space, we consider that the measurements included the types of air masses that contributed to the mixing. This leads to the next point.

3) Missing domain can lead to misleading interpretation – arrow 2a in Fig.11

Although this is an excellent dataset, it misses an important section of the system. The clouds on the south side of the jet blocked the lidar and the cross section did not cover the lower quadrant south of the intrusion, which is a region actively involved in mixing. Because of this, the interpretation of the data in tracer space needs to take into consideration not all source regions, or the types of air masses supplied the mixing lines, are represented by the measurements. In particular, the missing section would have represented the air mass type supplies "Mix-2". It would be more consistent with the picture of two layers involved in quasi-isentropic mixing: the layer above the jet core (PT > 340K, Mix-1), and the layer below the jet core (PT < 340 K, Mix-2). The presence of the cloud is indirect evidence of upward motion that can generate mixing against the downward intrusion.

Additionally, the data scatters shown in Figs. 9a-b support the strong chemical discontinuity at the high latitude tropopause. Figure 2 also show no evidence of upward mixing across the PV tropopause. Dynamically, this region is dominated by the generally downward motion in the ageostrophic flow behind the fold. These considerations

make the two-way arrow 2a in Fig 11 and the identification of "cross-isentropic transition", which suggests the path of mixing, conceptually problematic.

Similarly, it would be good to discuss the likely implication of missing domain to Fig. 7b. What would it look like if the leading edge of the fold is filled with the TRO3 type of air masses with higher moisture?

4) Take home massages in section 4

Figure 11 is a very good way of summarizing the emerging picture from the analysis (with the arguable point re process 2a). The section 4 overall can be strengthened by a set of distinct take home massages. It would be much more effective than the continued narratives, summarizing the previous sections. A few well distilled points would also help the readers digest the results, for example, which previous understanding or school of thoughts your analysis supports, verifies, or completes? What new questions or hypotheses this work has brought forward?

Minor suggestions for specifics:

1) Mention the WISE configuration is nadir viewing, although it becomes obvious later.

2) Give the flight altitude (FL450, approx. 14 km) for readers unfamiliar with aircraft work jargons.

3) It would be helpful to have the same map projection for Figs. 1 and 2, although you may need to zoom into different areas. This is especially confusing when the track orientations are different and lat/lon grids are not labeled.

4) The inverse ozone vertical gradient on the south side of the jet is confusing to me. It may deserve a few words of discussion: Is the increase to 100 ppb ozone criterion dictated by the high latitude portion of TRO-3?

5) The observation shown in Pan et al., 2006, although not the same system, is a very closely related work of lidar cross section and in situ tracer-tracer analysis in a similar

season and region. It probably should be cited as a contrast to what you have now more than just ozone.

Pan, L. L., P. Konopka, and E. V. Browell (2006), Observations and model simulations of mixing near the extratropical tropopause, J. Geophys. Res., 111, D05106, doi:10.1029/2005JD006480.

---

## Referee Comment (RC2) · Anonymous Referee #2 · 12 Dec 2020

Mixing at the extratropical tropopause as characterized by collocated airborne H2O and O3 lidar observations

Andreas Schäfler, Andreas Fix, Martin Wirth

This paper presents novel measurements of a 2-D cross-section of ozone and water vapour across a jet stream, identifying regions of mixed air using tracer-tracer distributions. There is considerable literature in this field drawn from in situ and satellite observations, and the methodology used in the paper follows previous work. The novelty therefore rests on the unique co-located lidar measurements that extend the previous studies into high-resolution 2-D curtains.

The study is certainly worthy of publication but I struggled with it in its present form. My main criticism is that the paper is much too long, with a wealth of detail presented at all turns that obscures the scientific argument. I sit necessary, for instance, to give a blow-by-blow description of each of the figures? Surely what is needed is to point out the features that feed into the scientific argument of the paper. As there is no conclusions section, the conclusions of the paper are included in the narrative, which means it is not clear what parts of the paper are necessary. It seems that the authors have applied all the analysis methods ever used in the past regardless of whether they bring new insight on the problem.

The difficulties start with the Introduction, which provides a comprehensive summary of previous work but does not identify the scientific problem that needs addressing – instead we are told '*For a more detailed characterization of the influence of the dynamics in individual synoptic situations on the distribution of trace gases in the ExTL, an observation capability for the instantaneous two-dimensional distribution of relevant trace gas along cross-sections through individual weather systems is required.*' Well yes, but why is a more detailed characterisation required? I'm not saying it isn't, I'm saying that clearly identifying the problem that needs to be addressed will help focus the remainder of the paper and allow un-necessary detail to be removed.

Section 2.2 is one of the areas that would benefit from considerable reduction in length. It reads like a review article on tracer-tracer correlations. What is the information that is relevant to the analysis in this paper?

What is the point of fig.4 and the accompanying text? Why not press straight on to the tracer-tracer correlations as these are the heart of the paper?

Again, I could not understand the point of fig.6 and the accompanying discussion – whereas fig 7a is a real highlight, comprehensively demonstrating the power of the combined lidar observations.

But 7b, the mixing factor analysis, doesn't seem to me to be telling us anything new. That may be because of the confusing way the diagnostic is described – e.g l.414-420 seem to be trying to say something important but whatever it is makes no sense to me. The authors should either remove this analysis altogether, or rewrite is much more clearly and concisely so that it comes to a conclusion that advances the scientific argument in the paper.

I thought fig.9 worked well, and followed on nicely from 5c and 7a.

What is the purpose of section 3.5, 'Isentropic trace gas gradients'? Is it to calculate the dynamical tropopause? If so, the section heading would be better as 'Comparison of the dynamical tropopause with isentropic trace gas gradients'. It seems that the key sentence in this section is that on l.511 '*The PV-gradient tropopause follows the regions of highest isentropic trace gas gradients much better than the 2-PVU isoline, especially in case of the O3 gradients, which confirms Kunz et al (2011b)*'. However, I don't agree with this sentence. First of all, the subjective 'much better' is not a scientific statement – by what criterion is it better, and how much better by this criterion? Secondly, the 2 PV contour in fig. 10f doesn't do too badly at all below 340 K. I don't think the tracer gradients support any particular definition of the dynamical tropopause, and challenge the authors to come up with a convincing diagnostic to show that they do.

The summary needs to set out more clearly what the conclusions of this work are, and how they advance our understanding of the tropopause region. There is a very good paper in here but the authors need to sharpen their arguments and dispense with superfluous detail in order to let it blossom.

Minor comments

l.16 'allows to identify' -> identifies (you can't use the construction '*subject* allows to *verb*' in English, it needs to be '*subject* allows *object* to *verb*'.) On line 150 I have suggested the latter construction; here and on l. 538 the 'allows to' is superfluous.

l.18 'surroundings'

l 18, 218 and 268 'indicative of'

l.20 'confirm that the mixing is strongest'

l.21 'the strongest'

l.23 'Although the methods do not allow conclusions to be drawn either on the individual mixing process or on the location and time of the event'

l.24 'allows the formation … T-T space to be discussed and hypotheses to be developed about mixing on ..'

l.25 'The 2-D lidar data presented'

l.49 'However, the DIAL they used was not capable of accurately measuring'

l.95 '..EXTL in the context of the'

l.125 'ratio of both signals as a function of the time taken to pass through the atmosphere, and knowledge about the exact absorption characteristics'

l.150 'allowed the mixing layer to be delineated'

l.200 what do you use as a definition of the dynamical tropopause? (I know you answer this question later but it helps the reader to know your definition at this point)

l 223 'tropospheric air'

l.229 'both in situ' (no comma)

l.234 'meridional' (not zonal)

l.338 *'high potential temperatures that increase towards lower ozone, which corresponds to vertically decreasing ozone values'* Fig. 3b shows a range of ozone values between 100 and 500 ppb for $\theta > 360$ K, which is not consistent with this sentence

l.386 This paragraph makes specific reference to the dynamical tropopause. How does this differ from the thermal tropopause well to the north of the jet?

l.406 'Please note……' This sentence does not make sense –please rewrite

l.41 surroundings

l.429 the callout '(fig 9b)' is in the wrong place

l.430 missing figure callout '()'

l.463 the sentence 'In dynamic…' has no verb in the second clause and does not make sense

l.486 what does 'angular' mean?

l.492 Below the jet maximum your PV tropopause is displaced towards lower, not higher PV

l.538 'allowed to identify' -> identified

l.539 'losing'

---

## Author Comment (AC1) · 16 Feb 2021

The comment was uploaded in the form of a supplement: https://acp.copernicus.org/preprints/acp-2020-1085/acp-2020-1085-AC1-supplement.pdf

[Figure]
**ACP 2020-185:** Mixing at the extratropical tropopause as characterized by collocated airborne H2O and O3 lidar observation

by Schäfler et al.

**Reply to review #1 by Laura Pan**

Dear Laura Pan, we are grateful for the positive review and the recognition of our work. We appreciate that you consider the manuscript to be unique and of high value and that it contributes to the state of art of this research topic. We thank you for the valuable suggestions and comments that helped us to improve our manuscript. Below, we answer each comment using a blue font. At the end we added an updated version using track changes. Please note that all references to lines refer to this revised version.

Transport and mixing near the extratropical tropopause associated with the jet dynamics, relationship between the meteorological background and the chemical distribution in the extratropical upper troposphere and lower stratosphere (ExUTLS) have been an active line of research for multiple decades. An important development in the past two decades is the use of tracer-tracer correlation. This line of work has been largely fueled by new observations from new satellite and especially research aircraft data. The analysis presented in this work expands the horizon using the new generation of airborne lidar observations that provided co-located ozone and water vapor measurements in a two-dimensional cross section intersects a region of active jet dynamics and stratospheric intrusion. The authors did an excellent analysis to connect the 2D ozone and water vapor observation with all major prior studies when limited in situ sampling or coarse satellite data were available. The analysis combining the tracer space diagram and geometric space distribution using this unique 2D data is a significant contribution to the topic of ExUTLS research, and it could serve as a road map for analyzing future observations in this layer. Given that the modeling community is working toward improving spatial resolutions of the chemical transport and/or chemistry climate models to resolve the tropopause region, the method demonstrated here can be applied readily for processsspecific model evaluations.

In summary, the new data presented is unique and of high value. The analysis presented contributes to the state of art on the topic of research. The large set of analysis details are presented in excellent clarity. I do have a few comments and suggestions below for the authors' consideration. Major comments and suggestions:

1)      It is too much of a missed opportunity not to include lapse rate tropopause in the analysis. An important question in the work of ExUTLS transport and STE is the relationship between various tropopause definitions, in particular, definitions of the dynamical tropopause and the temperature lapse rate based thermal tropopause from WMO. Chemical discontinuity and tracer-tracer relationship is a unique way to shed light on the physical behavior revealed by the relationship of the two definitions. The work did an excellent job diagnosing the dynamical tropopause using PV gradient versus fixed PV surface. It would be very valuable to diagnose the vertical discontinuity at the tropopause represented by the temperature lapse rate (discontinuity of the static stability) and PV contours, and the chemical mixing versus discontinuity associated with the conditions where they agree or show large separation. In this direction, it is worthwhile to present the Lapse rate tropopause on the cross section, and to show additional LRT relative profiles in Fig. 4. I understand that you don't have vertical legs for in situ measured temperature profiles but ERA5 300m resolution data should be able to provide a good approximation.

[Figure]

**Fig. 1.**

**Supplement:**

**ACP 2020-185:** Mixing at the extratropical tropopause as characterized by collocated airborne H2O and O3 lidar observation

by Schäfler et al.

**Reply to review #1 by Laura Pan**

Dear Laura Pan, we are grateful for the positive review and the recognition of our work. We appreciate that you consider the manuscript to be unique and of high value and that it contributes to the state of art of this research topic. We thank you for the valuable suggestions and comments that helped us to improve our manuscript. Below, we answer each comment using a blue font. At the end we added an updated version using track changes. Please note that all references to lines refer to this revised version.

Transport and mixing near the extratropical tropopause associated with the jet dynamics, relationship between the meteorological background and the chemical distribution in the extratropical upper troposphere and lower stratosphere (ExUTLS) have been an active line of research for multiple decades. An important development in the past two decades is the use of tracer-tracer correlation. This line of work has been largely fueled by new observations from new satellite and especially research aircraft data. The analysis presented in this work expands the horizon using the new generation of airborne lidar observations that provided co-located ozone and water vapor measurements in a two-dimensional cross section intersects a region of active jet dynamics and stratospheric intrusion. The authors did an excellent analysis to connect the 2D ozone and water vapor observation with all major prior studies when limited in situ sampling or coarse satellite data were available. The analysis combining the tracer space diagram and geometric space distribution using this unique 2D data is a significant contribution to the topic of ExUTLS research, and it could serve as a road map for analyzing future observations in this layer. Given that the modeling community is working toward improving spatial resolutions of the chemical transport and/or chemistry climate models to resolve the tropopause region, the method demonstrated here can be applied readily for processspecific model evaluations.

In summary, the new data presented is unique and of high value. The analysis presented contributes to the state of art on the topic of research. The large set of analysis details are presented in excellent clarity. I do have a few comments and suggestions below for the authors' consideration. Major comments and suggestions:

1) It is too much of a missed opportunity not to include lapse rate tropopause in the analysis. An important question in the work of ExUTLS transport and STE is the relationship between various tropopause definitions, in particular, definitions of the dynamical tropopause and the temperature lapse rate based thermal tropopause from WMO. Chemical discontinuity and tracer-tracer relationship is a unique way to shed light on the physical behavior revealed by the relationship of the two definitions. The work did an excellent job diagnosing the dynamical tropopause using PV gradient versus fixed PV surface. It would be very valuable to diagnose the vertical discontinuity at the tropopause represented by the temperature lapse rate (discontinuity of the static stability) and PV contours, and the chemical mixing versus discontinuity associated with the conditions where they agree or show large separation. In this direction, it is worthwhile to present the Lapse rate tropopause on the cross section, and to show additional LRT relative profiles in Fig. 4. I understand that you don't have vertical legs for in situ measured temperature profiles but ERA5 300m resolution data should be able to provide a good approximation.

We followed your suggestion and added the location of the thermal tropopause to the lidar cross section in the revised Fig. 7 (now Fig. 6). Following the recommendation of reviewer #2 we decided to remove Fig. 4 and not to show additional LRT profiles. We think that Fig. 4 does not add significant detail and was rather a bridge to T-T space topic. However, we kept all relevant information.

The location confirms earlier findings where the tropopause is flat and we also find the typical split near the jet stream. Therefore, we added (see L425f): *"In the northern part of the flight section, the bottom of the ExTL (MIX-2) agrees with the dynamical tropopause while the thermal tropopause lies within the ExTL and approximately follows the 3.5 PVU contour. Please note that the thermal tropopause provides a typical split structure near the jet stream while the dynamical tropopause proceeds vertically (e.g. Randel et al., 2007)."*

2) It is important to have a clear message that mixing processes that created these mixed regions did not happen in the observed cross section, but rather, upstream. As we draw connections between the air masses in the observed cross-section and identify the sources of mixing in the tracer space, we consider that the measurements included the types of air masses that contributed to the mixing. This leads to the next point.

This is an important and valid comment and exactly what we intended to say! It was our idea to show that mixing lines even form for data subsets that are not linked via quasi-instantaneous mixing. We hope this becomes clear from our completely revised discussion of the formation of mixing lines and their interpretation (please see L622ff and next comment).

3) Missing domain can lead to misleading interpretation – arrow 2a in Fig.11 Although this is an excellent dataset, it misses an important section of the system. The clouds on the south side of the jet blocked the lidar and the cross section did not cover the lower quadrant south of the intrusion, which is a region actively involved in mixing. Because of this, the interpretation of the data in tracer space needs to take into consideration not all source regions, or the types of air masses supplied the mixing lines, are represented by the measurements. In particular, the missing section would have represented the air mass type supplies "Mix-2". It would be more consistent with the picture of two layers involved in quasi-isentropic mixing: the layer above the jet core (PT > 340K, Mix-1), and the layer below the jet core (PT < 340 K, Mix-2). The presence of the cloud is indirect evidence of upward motion that can generate mixing against the downward intrusion. Additionally, the data scatters shown in Figs. 9a-b support the strong chemical discontinuity at the high latitude tropopause. Figure 2 also show no evidence of upward mixing across the PV tropopause. Dynamically, this region is dominated by the generally downward motion in the ageostrophic flow behind the fold. These considerations make the two-way arrow 2a in Fig 11 and the identification of "cross-isentropic transition", which suggests the path of mixing, conceptually problematic. Similarly, it would be good to discuss the likely implication of missing domain to Fig. 7b. What would it look like if the leading edge of the fold is filled with the TRO3 type of air masses with higher moisture?

We agree that a remark on the missing data in the lower part of the fold was missing. We added (L644ff): "Although the mixing factor provides some indication of increased isentropic mixing below the jet stream, unfortunately no observations are available in the lowest part of the tropopause fold due to clouds. However, a separated mixing regime at the northern edge of the fold suggests recent mixing between ExTL and moist tropospheric air beneath."

We also have noticed that the arrows in former Fig. 11 can easily be misinterpreted. We meant them to represent transitions that form mixing lines even in the situations that clearly do not represent recent or quasi-instantaneous mixing. Therefore, we decided to remove Fig. 11 and leave it with a more

precise discussion of this issue (L622ff). We think that this analysis is one of the major contributions of this paper and would be glad to get further feedback whether this became clearer.

4) Take home massages in section 4

Figure 11 is a very good way of summarizing the emerging picture from the analysis (with the arguable point re process 2a). The section 4 overall can be strengthened by a set of distinct take home massages. It would be much more effective than the continued narratives, summarizing the previous sections. A few well distilled points would also help the readers digest the results, for example, which previous understanding or school of thoughts your analysis supports, verifies, or completes? What new questions or hypotheses this work has brought forward?

Although you consider Fig. 11 a good way of summarizing the results we decided to remove it (see comment before). We completely revised Sect. 4 (also following the recommendation of reviewer #2). After a short summary the discussion now follows research questions that we added to the introduction. However, we found it difficult to add a list of short distilled take-home message, but tried our best to be more specific regarding the outcome of this paper and how this is related to the current understanding. Please see revised Section 4.

Minor suggestions for specifics:

1) Mention the WISE configuration is nadir viewing, although it becomes obvious later.

We added the following sentence to Sect. 2.1 (see L149): "However, during WISE the lidar was exclusively measuring nadir."

2) Give the flight altitude (FL450, approx. 14 km) for readers unfamiliar with aircraft work jargons.

Done.

3) It would be helpful to have the same map projection for Figs. 1 and 2, although you may need to zoom into different areas. This is especially confusing when the track orientations are different and lat/lon grids are not labeled.

We changed the map projection in Fig. 1 to obtain a comparable orientation of the track. We kept the large area which is important to understand the transport related to the upstream trough. Additionally, the grid was labelled in Fig. 1a and b.

4) The inverse ozone vertical gradient on the south side of the jet is confusing to me. It may deserve a few words of discussion: Is the increase to 100 ppb ozone criterion dictated by the high latitude portion of TRO-3?

We were not completely sure where in the manuscript you exactly refer to. In Sect. 3.2 we say "In the troposphere,  $O_3$  is comparatively low (20–100 ppb) with the lowest values occurring in the midtropospheric moist air to the south of the jet stream being indicative of recent transport from the lower troposphere." This was supported by the distribution in T-T space (Sect. 3.3) and the air mass classification. The above located air most-likely originates from the tropics which we conclude from the low moisture content. We also discuss the role of some in-mixing at highest altitudes in Sect. 4.

5) The observation shown in Pan et al., 2006, although not the same system, is a very closely related work of lidar cross section and in situ tracer-tracer analysis in a similar season and region. It probably should be cited as a contrast to what you have now more than just ozone.

Pan, L. L., P. Konopka, and E. V. Browell (2006), Observations and model simulations of mixing near the extratropical tropopause, J. Geophys. Res., 111, D05106,

Thanks for pointing to this important paper. We added it to the introduction ("Pan et al. (2006) combined lidar O3 
[revised manuscript text omitted]
 aThe synoptic situation of a text-book-like situation with a transect of a zonal extratropical jet stream observed-over the North Atlantic Ocean during WISE-on 1 October 2017 (is explained in Sect. 3.1). In Sect. 3.2 to 3.4 the lidar data is presented along cross-sections, in T-T space and in combined view, respectively. The interrelation of chemical and dynamical discontinuities at the midlatitude tropopause is described in Sect. 3.5. A discussion of the
- 150 results and conclusion is given in Sect. 4.

For the first time, distinct mixing regimes for a range of isentropic levels across the tropopause allow a detailed depiction and description of the transition from stratosphere to troposphere (Sect. 3.2–3.4). Additionally, isentropic O3 and H2O gradients are determined to investigate the interrelation of chemical and dynamical discontinuities at the mid latitude tropopause (Sect. 3.5). Section 4 gives a summary of the results. In a follow up paper we aim to apply Lagrangian diagnostics to investigate the role of

15 differing transport pathways and the timescales that were relevant for the complex distribution of O3 and H2O in this case.

**2 Data and methods**

**2.1 Lidar observations onboard HALO**

[revised manuscript text omitted]

A prerequisite for the T-T method is that the distributions are controlled by transport processes i.e. that the lifetime of the used trace gases is longer than the timescale of transport and mixing at the tropopause, which is in the order of weeks Different trace gases were used: for example, carbon monoxide (CO) as tropospheric tracer in combination with Q2 as stratospheric tracer (e.g. Hoor et al., 2002). In this study O3-H2O correlations are applied. Pan et al. (2007) discuss the applicability of O3-H2O correlations and note

- that H2O is a suitable tropospheric tracer despite the fact that it is not perfectly long-lived as phase changes may cause non-220 conservation of the gas phase H2O concentration. As discussed in Hegglin et al. (2009), the exponential decrease of H2O across the tropopause makes it a very useful source of information about transport into the stratosphere as even small amounts of H2O become visible as signature of increased H2O. In the stratosphere, methane oxidation can produce H2O which is, however, rather small in the LS and therefore often neglected (Pan et al., 2014b). Due to the large dynamic range of H2O of four orders of magnitude from
- 225 the troposphere to the stratosphere, T-T depictions use the H2O data is displayed in allogarithmic scaling for H2O-to be able to distinguish the mixing lines (e.g., Hegglin et al., 2009; Tilmes et al., 2010), which are typically curved in lin-log T-T diagrams, more easily.

[revised manuscript text omitted]

---

## Author Comment (AC2) · 16 Feb 2021

We thank the reviewer for the constructive comments and thorough review. We are glad that you considered our manuscript worthy for publication. We tried to better specify the scientific argument and the discussion of our results. Below you find detailed response (in blue font) to all your comments. At the end we added an updated version using track changes. Please note that all references to lines refer to this revised track change version.

This paper presents novel measurements of a 2-D cross-section of ozone and water vapour across a jet stream, identifying regions of mixed air using tracer-tracer distributions. There is considerable literature in this field drawn from in situ and satellite observations, and the methodology used in the paper follows previous work. The novelty therefore rests on the unique co-located lidar measurements that extend the previous studies into high-resolution 2-D curtains.

The study is certainly worthy of publication but I struggled with it in its present form. My main criticism is that the paper is much too long, with a wealth of detail presented at all turns that obscures the scientific argument. I sit necessary, for instance, to give a blow-by-blow description of each of the figures? Surely what is needed is to point out the features that feed into the scientific argument of the paper. As there is no conclusions section, the conclusions of the paper are included in the narrative, which means it is not clear what parts of the paper are necessary. It seems that the authors have applied all the analysis methods ever used in the past regardless of whether they bring new insight on the problem.

We changed the original version at several places to improve the readability and to follow your recommendations: This reduced the overall length by about 3 pages. In the introduction we added research questions to clarify the scientific focus of the study. Throughout the manuscript we removed conclusions from the narrative and we also removed two figures. As the other reviewer positively mentioned the clarity of the analysis, we kept the overall structure and tried to steer a middle course between keeping enough detail and shortening some descriptions. After a short summary, the discussion of the results evolves from the research question. Please see detailed comments below.

We agree that this study follows previous work and applies established methods to a novel set of lidar data, which is explicitly stated in the introduction and discussion. Although we do not propose new methods here, we think that the results are unique in the sense that the allow a verification and novel visualization of established concepts that are based on limited observation types and/or model simulations. We aimed at clarifying this.

The difficulties start with the Introduction, which provides a comprehensive summary of previous work but does not identify the scientific problem that needs addressing – instead we are told 'For a more detailed characterization of the influence of the dynamics in individual synoptic situations on the distribution of trace gases in the ExTL, an observation capability for the instantaneous two-dimensional distribution of relevant trace gas along cross-sections through individual weather systems is required.' Well yes, but why is a more detailed characterisation required? I'm not saying it isn't, I'm saying that clearly identifying the problem that needs to be addressed will help focus the remainder of the paper and allow un-necessary detail to be removed.

We revised the introduction and included several research questions. We hope that it becomes clear how our case study wants to contribute to a more detailed characterization of the ExTL related to the dynamical situation, which was the aim of the WISE field experiment. The formulation "*For a…*" was removed.

Section 2.2 is one of the areas that would benefit from considerable reduction in length. It reads like a review article on tracer-tracer correlations. What is the information that is relevant to the analysis in this paper?

We followed the recommendation and revised Sect. 2.2 leading to a reduced length of 1.5 pages (instead of 2). We removed not relevant details on the ExTL statistical investigations and moved some content to Sect. 4. Some off-topic sentences were removed. However, we still consider the description of the basic methods, the appearance of mixing lines in T-T space, the applicability of $O_3$ and $H_2O$ and the selection methods of ExTL mixed air as relevant to understand our approach and the results.

What is the point of fig.4 and the accompanying text? Why not press straight on to the tracer-tracer correlations as these are the heart of the paper?

We removed Fig. 4 and the accompanying text.

Again, I could not understand the point of fig.6 and the accompanying discussion – whereas fig 7a is a real highlight, comprehensively demonstrating the power of the combined lidar observations.

We decided to keep Fig. 6 (now Fig. 5) as it gives context to the observations in T-T space that is considered to be important for the overall understanding of the observations in both depictions.

But 7b, the mixing factor analysis, doesn't seem to me to be telling us anything new. That may be because of the confusing way the diagnostic is described – e.g l.414-420 seem to be trying to say something important but whatever it is makes no sense to me. The authors should either remove this analysis altogether, or rewrite is much more clearly and concisely so that it comes to a conclusion that advances the scientific argument in the paper.

We think that the mixing factor diagnostic gives valuable information about the composition of the ExTL through visualization of the transition therein. In combination with the T-T plot it informs about the strength and rapidity of the transition which is the motivation for the detailed analysis of data subsets in Fig. 9 (former Fig. 10, see also introduction to Sect. 3.5). We followed your recommendation and removed Fig. 8b (and related text in l. 414-420). We merged former Fig. 7b and 8a to a new Fig. 7 and tried to improve the related discussion.

I thought fig.9 worked well, and followed on nicely from 5c and 7a. Thanks!

What is the purpose of section 3.5, 'Isentropic trace gas gradients'? Is it to calculate the dynamical tropopause? If so, the section heading would be better as 'Comparison of the dynamical tropopause with isentropic trace gas gradients'. It seems that the key sentence in this section is that on l.511 'The PV-gradient tropopause follows the regions of highest isentropic trace gas gradients much better than the 2-PVU isoline, especially in case of the O3 gradients, which confirms Kunz et al (2011b)'. However, I don't agree with this sentence. First of all, the subjective 'much better' is not a scientific statement – by what criterion is it better, and how much better by this criterion? Secondly, the 2 PV contour in fig. 10f doesn't do too badly at all below 340 K. I don't think the tracer gradients support any particular definition of the dynamical tropopause, and challenge the authors to come up with a convincing diagnostic to show that they do.

As shown in earlier studies the interpretation of transport and mixing processes in the ExTL depend on the definition of the tropopause. Therefore, we investigate the relationship between chemical and

*dynamical discontinuities which is not possible without curtain-like observations. We realized that this was not treated in the introduction which we improved by better motivating the topic. We follow the recommendation and changed the title of Sect 3.5 to "Chemical and dynamical discontinuities at the tropopause".*

*"much better" was removed and we tried to better describe our interpretation of the relative location of trace gas and PV-gradient tropopause. We agree with your opinion about the situation below 340 K. However, above, the strongest gradients (especially of the stratospheric tracer $O_3$) is shifted towards higher PV and is better represented by the PV-gradient tropopause. This reads now as (L566): "The PV-gradient tropopause better represents the regions of highest isentropic trace gas gradients than the 2-PVU isoline, especially in case of the $O_3$ gradients. It follows the center of maximum $O_3$ gradients at the level of highest wind speeds and above. Maximum $H_2O$ gradients are located north of the PV-gradient tropopause at even higher PV values at the jet stream level where pronounced gradients are visible."*

*This is the first observational evidence of this relationship at the extratropical tropopause. We added some details to the summary and discussions.*

The summary needs to set out more clearly what the conclusions of this work are, and how they advance our understanding of the tropopause region. There is a very good paper in here but the authors need to sharpen their arguments and dispense with superfluous detail in order to let it blossom.

*We agree to your comment that the conclusion wasn't clear enough, which was partly connected to missing clarity regarding the scientific questions that we added now. This is also in line with reviewer #1 and therefore we completely revised Sect. 4. The restructured paragraphs now follow the questions that were posed in the introduction. As stated above we moved some conclusion from the main text to the end and tried to make clear how our analysis confirmed earlier concepts and how we contribute to a better understanding of the tropopause region. Please re-read the revised version of Sect. 4*

Minor comments

l.16 'allows to identify' -> identifies (you can't use the construction 'subject allows to verb' in English, it needs to be 'subject allows object to verb'.) On line 150 I have suggested the latter construction; here and on l. 538 the 'allows to' is superfluous. *Done.*

l.18 'surroundings' *Done.*

l 18, 218 and 268 'indicative of' *Done*

l.20 'confirm that the mixing is strongest' *Done*

l.21 'the strongest' *Done*

l.23 'Although the methods do not allow conclusions to be drawn either on the individual mixing process or on the location and time of the event' *The sentence was changed accordingly.*

l.24 'allows the formation … T-T space to be discussed and hypotheses to be developed about mixing on ..' *Done*

l.25 'The 2-D lidar data presented' *Done*

l.49 'However, the DIAL they used was not capable of accurately measuring' *Done*

l.95 '..EXTL in the context of the' *This sentence was removed.*

l.125 'ratio of both signals as a function of the time taken to pass through the atmosphere, and knowledge about the exact absorption characteristics' Done

l.150 'allowed the mixing layer to be delineated' Done

l.200 what do you use as a definition of the dynamical tropopause? (I know you answer this question later but it helps the reader to know your definition at this point)

We changed the sentence to (L246): *"The main parameters of interest are wind speed to identify the jet stream, pressure and potential temperature for the vertical context as well as potential vorticity (PV). Please note that 2 PVU are used for locating the dynamical tropopause."*

l 223 'tropospheric air Done

l.229 'both in situ' (no comma) Done

l.234 'meridional' (not zonal) Done

l.338 'high potential temperatures that increase towards lower ozone, which corresponds to vertically decreasing ozone values' Fig. 3b shows a range of ozone values between 100 and 500 ppb for θ > 360 K, which is not consistent with this sentence

Maybe this is a misunderstanding. The sentence refers to air mass STRA only which shows decreased Theta at the lower boundary (~300 ppb). An explanation is given in the subsequently following sentence.

l.386 This paragraph makes specific reference to the dynamical tropopause. How does this differ from the thermal tropopause well to the north of the jet?

We added the thermal tropopause to Fig. 6 and Fig. 7 and added: *"In the northern part of the flight section, the bottom of the ExTL (MIX-2) agrees with the dynamical tropopause while the thermal tropopause lies within the ExTL and approximately follows the 3.5 PVU contour. Please note that the thermal tropopause provides a typical split structure near the jet stream while the dynamical tropopause proceeds vertically (e.g. Randel et al., 2007)."*

l.406 'Please note......' This sentence does not make sense –please rewrite

Changed to: *"Please note that the selected ExTL air masses slightly differ from Fig. 4 due to the application of constant minimum thresholds."*

l.41 surroundings Done

l.429 the callout '(fig 9b)' is in the wrong place Corrected

l.430 missing figure callout '()' Removed

l.463 the sentence 'In dynamic…' has no verb in the second clause and does not make sense. This section was revised and the sentence was deleted.

l.486 what does 'angular' mean? Changed to (L541): *"meridian resulted in edged PV distributions on isentropes"*

l.492 Below the jet maximum your PV tropopause is displaced towards lower, not higher PV. We removed this sentence

l.538 'allowed to identify' -> identified Done

l.539 'losing' Done

[revised manuscript text omitted]